# Unusual Sabatier principle on high entropy alloy catalysts for hydrogen evolution reactions

Zhi Wen Chen [1,2,6], Jian Li [1,6], Pengfei Ou [3], Jianan Erick Huang [3], Zi Wen [1], LiXin Chen [2], Xue Yao [2], GuangMing Cai [4], Chun Cheng Yang [1,7] ✉, Chandra Veer Singh [2,5,7] ✉ & Qing Jiang [1,7] ✉

The Sabatier principle is widely explored in heterogeneous catalysis, graphically depicted in volcano plots. The most desirable activity is located at the peak of the volcano, and further advances in activity past this optimum are possible by designing a catalyst that circumvents the limitation entailed by the Sabatier principle. Herein, by density functional theory calculations, we discovered an unusual Sabatier principle on high entropy alloy (HEA) surface, distinguishing the "just right" ($\Delta G_{H^*} = 0$ eV) in the Sabatier principle of hydrogen evolution reaction (HER). A new descriptor was proposed to design HEA catalysts for HER. As a proof-of-concept, the synthesized PtFeCoNiCu HEA catalyst endows a high catalytic performance for HER with an overpotential of 10.8 mV at −10 mA cm$^{-2}$ and 4.6 times higher intrinsic activity over the state-of-the-art Pt/C. Moreover, the unusual Sabatier principle on HEA catalysts can be extended to other catalytic reactions.

In the Sabatier principle, the adsorbate should bind neither too weakly (lest reactants fail to activate) nor too strongly (lest products fail to dissociate)[1,2]. It provides useful guidance in heterogeneous catalysis and is also held up as a rule or limit to be circumvented when one seeks further to advance catalytic performance[3–8]. The resultant volcano plots have been used to guide catalyst design for the $CO_2$ reduction reaction ($CO_2$RR)[9], nitrogen reduction reaction (NRR)[10], hydrogen evolution reaction (HER)[11], and oxygen reduction/evolution reaction (ORR/OER)[12,13]. "Just right" adsorption energy is the pursuit of all chemical reactions. However, although we achieve the "just right" adsorption energy, the catalytic activity is just infinitely close to the peak. Further breakthroughs or over the volcano in catalytic activity are almost impossible. For instance, Nørskov et al. applied high-throughput density functional theory (DFT) calculations to screen out BiPt with the optimal adsorption energy of H* among 736 alloy

systems. This alloy was synthesized and tested experimentally and showed improved HER performance compared with Pt, however, still below the volcano peak[14].

Circumventing the volcano relationship is plausible to achieve a breakthrough or over the volcano in catalytic activity[15], and numerous efforts have been focused on this issue[8,16,17]. For example, Chen et al. demonstrated that the volcano relationship can be broken by building an interface between transition metals and LiH[18]. Unfortunately, the relatively few active sites have impeded the wide application of interface catalysis[19]. Another strategy of strain effect was proposed by Khorshidi et al., the surface strain has to occur either externally by applying mechanical loading or internally by creating complex core-shell structures or interfaces[8]. Another potential phenomenon that could circumvent the volcano relationship is H* spillover, in which H* exhibits the ability to diffuse between different active sites[20–22]. Dai

[1]Key Laboratory of Automobile Materials (Jilin University), Ministry of Education, and School of Materials Science and Engineering, Jilin University, Changchun 130022, China. [2]Department of Materials Science and Engineering, University of Toronto; 184 College Street, Suite 140, Toronto, ON M5S 3E4, Canada. [3]Department of Electrical and Computer Engineering, University of Toronto, Toronto, ON M5S 1A4, Canada. [4]Department of Chemical Engineering and Applied Chemistry, University of Toronto; 200 College Street, Toronto, ON M5S 3E5, Canada. [5]Department of Mechanical and Industrial Engineering, University of Toronto; 5 King's College Road, Toronto, ON M5S 3G8, Canada. [6]These authors contributed equally: Zhi Wen Chen, Jian Li. [7]These authors jointly supervised this work: Chun Cheng Yang, Chandra Veer Singh, Qing Jiang. ✉e-mail: ccyang@jlu.edu.cn; chandraveer.singh@utoronto.ca; jiangq@jlu.edu.cn

et al. reported a single-phase complex oxide La$_2$Sr$_2$PtO$_{7+\delta}$ as a high performance HER electrocatalyst utilizing an atomic-scale H* spillover effect between multifunctional catalytic sites[23]. High entropy alloys (HEAs) with huge composition space have complex surface active sites, resulting in spatially varying adsorption of intermediates[24]. Some active sites with strong adsorption can be used to activate reactants, while some active sites with weak adsorption are favorable for the formation of products, which circumvents the Sabatier principle if the intermediates can easily diffuse on the HEA surface. It means the HEA catalysts provide new opportunities to achieve a breakthrough over the volcano in catalytic activity.

In this article, we found that HEA surface with spatially varying adsorption free energy of H* ($\Delta G_{H*}$) circumvents the Sabatier principle of HER. DFT calculations indicate that the $\Delta G_{H*}$ on HEA catalysts follows a Gaussian distribution [$X \sim N(\mu, \sigma^2)$, $\mu$: expectation; $\sigma$: standard deviation] due to the gradient electron distribution and the diffusion of H* on HEA surface is fairly easy with the small barrier of 0.232 eV. Some sites with strong adsorption ($\Delta G_{H*} < \mu - \sigma$) are used for H* adsorption, while some sites with weak adsorption ($\Delta G_{H*} > \mu + \sigma$) are used for H$_2$ formation. It means that the catalytic activity for HER will be better if the $\mu$ is closer to 0 eV and the $\sigma$ is larger, which is defined as an unusual Sabatier principle. The $\mu$ and $\sigma$ values could be regulated by the composition, strain effects, and synthesis conditions of HEA catalysts. Guided by these theoretical findings, a PtFeCoNiCu catalyst with electron and composition gradients has been precisely fabricated, exhibiting excellent catalytic performance with an overpotential of 10.8 mV at −10 mA cm$^{-2}$ and more than four times higher intrinsic activity over the state-of-the-art Pt/C. We also show that this unusual Sabatier principle can be extended to other adsorbents (C*, O*, and N*) on HEA surfaces, indicating potential applications for variety of catalytic reactions.

## Results

### Dual gradient PtFeCoNiCu HEA model
A PtFeCoNiCu HEA catalyst was originally designed based on the following three aspects: (1) Pt catalyzes HER and is a good choice of the active site for HER[14]; (2) the smaller atomic radii of Fe (1.56 Å), Co (1.52 Å), Ni (1.49 Å), and Cu (1.45 Å) than that of Pt (1.77 Å) would produce compressive strain on the HEA surface, resulting in a weaker H* adsorption on surface Pt sites, which promotes the catalytic performance for HER[25,26]; and (3) the cost of catalysts could be greatly reduced by using non-noble metals[27].

Taking into consideration that (i) some metals (Fe, Co, Ni, Cu) will be corroded away in an acidic electrolyte during the HER process, and (ii) the outer atomic layers will be etched more seriously than the inner layers, a PtFeCoNiCu HEA model with a Pt concentration gradient of 100.0 %, 50.0 %, 25.0 %, 12.5 %, and 12.5 % for the five layers has been designed, as shown in Fig. 1a. The coordination atoms to surface Pt active sites are diverse due to the nature of HEA, which results in various electron redistributions (electronic gradient), as shown in Fig. 1b. The corresponding Bader charge distribution is described in Supplementary Fig. 1. Such an electronic gradient causes different adsorption abilities for H*, where surface Pt sites with strong H* adsorption are active centers for the Volmer reaction (* + H$^+$ + e$^-$ → H*) while the ones with weak H* adsorption are active centers for the Tafel (H* + H* → H$_2$) or Heyrovsky reaction (H* + H$^+$ + e$^-$ → H$_2$).

Fe, Co, Ni, and Cu with smaller atomic radii than that of Pt will induce a compressive strain on the surface Pt atoms, which further regulates the electron structures of active sites[25]. Figure 1c shows that the energy level of $d$ orbitals in surface Pt atoms is gradually away from the Fermi level with increasing compressive strain, indicating the diminished activity. This phenomenon can be further quantified by their $d$-band center ($\varepsilon_d$) values, where a more negative $\varepsilon_d$ value indicates a weaker adsorption ability[28]. With the increase of compressive strain, the $\varepsilon_d$ values change from −1.66 eV (HEA without strain) to −1.72,

−1.75, −1.89, and −2.03 eV for 1.4%-HEA (HEA with 1.4% compressive strain), 3.2%-HEA, 5.0%-HEA, and 6.8%-HEA, respectively. The composition-strain-$\varepsilon_d$-activity relation allows for designing HEA catalysts with optimal adsorption energy via composition regulation[25].

### Gaussian distribution of $\Delta G_{H*}$ on PtFeCoNiCu HEA
$\Delta G_{H*}$ was calculated on the designed HEA (111) with different strains (see Fig. 1d and Supplementary Fig. 2), including more than 400 datapoints. The $\Delta G_{H*}$ distribution roughly conforms to the Gaussian distribution [$X \sim N(\mu, \sigma^2)$]. Herein, $\mu$ and $\sigma^2$ determine the location and the variance of $\Delta G_{H*}$, respectively. As shown in Fig. 1d, the $\mu$ value increases with increasing compressive strain. This is consistent with the $d$-band center theory, where a larger compressive strain brings more negative $\varepsilon_d$, resulting in weaker adsorption[28,29]. The corresponding structure (strain)-property ($\varepsilon_d$)-performance ($\mu$) relation is shown in Supplementary Fig. 3. Note that the compressive strain shows little influence on the $\sigma$ value. A larger $\sigma$ value indicates that some adsorption sites have stronger adsorptions while other adsorption sites have weaker adsorptions, which requires a larger electronic gradient on the surface. Above all, the two parameters ($\mu$ and $\sigma$) in the Gaussian distribution of $\Delta G_{H*}$ could be regulated by the type and number of alloying elements in HEA, which bring various strains and surface electronic gradients.

As is well known, $\Delta G_{H*}$ = 0 eV denotes the optimal catalytic performance of catalysts for HER based on the Sabatier principle[11]. Different from the traditional bimetallic and ternary metallic catalysts[14,26,30], the active sites of HEA are more diverse and their $\Delta G_{H*}$ values follow a Gaussian distribution, rather than a definite value. Hence, the Sabatier principle and the criterion of $\Delta G_{H*}$ = 0 eV are no longer efficient for HEA catalysts. In this work, we proposed an unusual Sabatier principle, where the Gaussian distribution of $\Delta G_{H*}$ with a $\mu$ value closer to 0 eV and a larger $\sigma$ value on HEA catalysts could be used as the descriptor of the higher catalytic activity for HER. Theoretically, the sites with $\Delta G_{H*} < \mu - \sigma$ and $\Delta G_{H*} > \mu + \sigma$ serve as the active centers for Volmer and Tafel (or Heyrovsky) reactions (see Supplementary Fig. 4), respectively. The larger $\sigma$ value indicates that the active center for the Volmer reaction has a stronger H* adsorption while the active center for Tafel (or Heyrovsky) reaction has a weaker H* adsorption (see Supplementary Fig. 5). This means that a larger $\sigma$ value results in faster Volmer and Tafel (or Heyrovsky) reactions, indicating a higher catalytic activity for HER. Moreover, the symmetry of the Gaussian distribution dictates that these two active centers are guaranteed to be the strongest and the weakest, respectively, only if $\mu$ = 0 eV (see Supplementary Fig. 6). Meanwhile, other sites with moderate $\Delta G_{H*}$ ($\mu - \sigma < \Delta G_{H*} < \mu + \sigma$) are the diffusion region (DR). The diffusion of H* on the HEA surfaces is known as H* spillover, which will be discussed in detail below.

### Reaction mechanism of HER on PtFeCoNiCu HEA
The 5.9%-HEA system was used as an example for studying the H* spillover based on the new descriptor of Gaussian distribution of $\Delta G_{H*}$ with the preferable $\mu$ = −0.034 eV and $\sigma$ = 0.041 eV (see Supplementary Fig. 7). The $\Delta G_{H*}$ values on some adsorption sites (48 sites, including three types of adsorption sites, as shown in Supplementary Fig. 8) of 5.9%-HEA are shown in Fig. 2a, where the green area denotes the DR. The active center for the Volmer reaction has the smallest $\Delta G_{H*}$ of −0.099 eV and the active center for the Tafel or Heyrovsky reaction has the largest $\Delta G_{H*}$ of 0.075 eV.

Both Volmer−Heyrovsky (V-H) and Volmer−Tafel (V-T) mechanisms in HER are considered on Pt (111) and 5.9%-HEA (111), as depicted in Fig. 2b, c. For the V-H mechanism on Pt (111), the potential limiting step (PLS) is the Heyrovsky step with $\Delta G_{Hey}$ = 0.375 eV, an endothermic reaction with the overpotential of 0.375 V. However, no PLS exists in the V-H mechanism on 5.9%-HEA (111) when considering the H* spillover. Both the Volmer and Heyrovsky steps are exothermic reactions

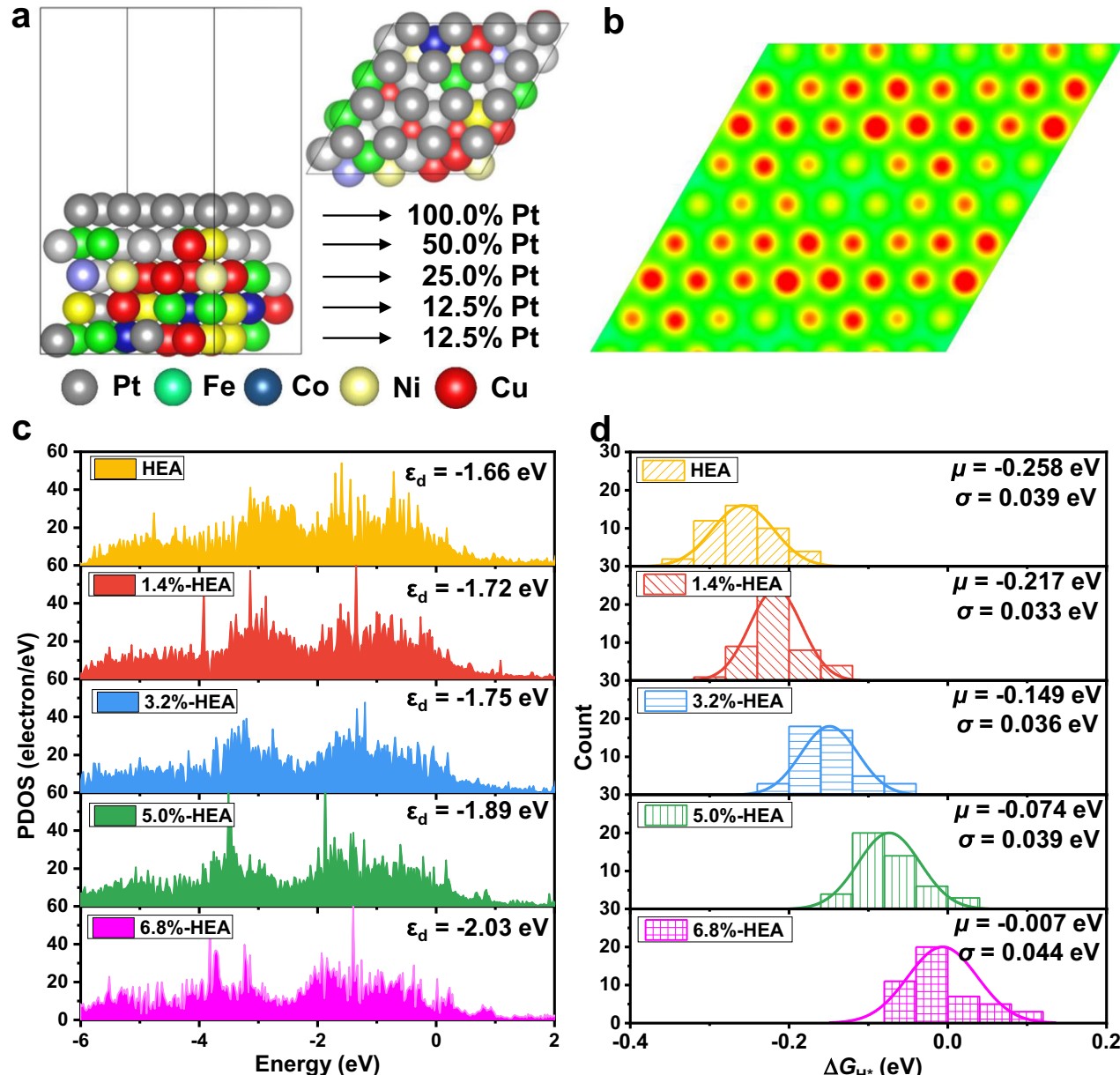

**Fig. 1 | DFT calculation of PtFeCoNiCu HEA model. a** The HEA (111) structure with Pt concentration gradient. **b** The electron localization function (ELF) contour for HEA (111) plane. The red areas represent rich electron distributions. **c** Partial density of states (PDOS) of $d$-band on surface Pt atoms in HEAs with different strains. The Fermi level is set to be 0 and the vertical dashed line is the $d$-band center ($\varepsilon_d$). **d** Gaussian distribution of adsorption free energy of H* ($\Delta G_{H^*}$) with the corresponding $\mu$ and $\sigma$ on HEAs with different strains.

with $\Delta G_{Vol-1} = -0.099$ eV and $\Delta G_{Hey} = -0.075$ eV, respectively. The adsorbed H* diffuses from Site A to Site B through the DR1 (see Fig. 2d) with the maximum energy barrier of 0.124 eV, which is much smaller than the energy barrier leading to a reaction rate of about 1 site$^{-1}$s$^{-1}$ at room temperature[31], indicating the exceedingly fast diffusion of H*.

For the V-T mechanism, the first two Volmer steps are exothermic reactions ($\Delta G_{Vol-1} = -0.375$ eV, $\Delta G_{Vol-2} = -0.201$ eV) on Pt (111). They are also exothermic reactions ($\Delta G_{Vol-1} = -0.099$ eV, $\Delta G_{Vol-2} = -0.091$ eV) on 5.9%-HEA (111). The following Tafel step has a large energy barrier of 1.128 eV on Pt (111). Although the energy barrier decreases to 0.466 eV with increasing H* coverage (see Supplementary Fig. 9), it is still much larger than that on 5.9%-HEA (111) (0.297 eV). For the V-T mechanism, DR2 is involved during the reaction process, as shown in Fig. 2e. The maximum energy barrier in DR2 is 0.232 eV, which is smaller than the rate determining step (RDS) of the Tafel reaction (0.297 eV) on 5.9%-

HEA (111). Above all, the H* spillover processes on both DRs wouldn't be the PLS or RDS during HER on 5.9%-HEA (111). To eliminate any potential influence arising from specific active centers, we randomly selected an additional active center on the 5.9%-HEA (111) and calculated the catalytic performance for HER, as shown in Supplementary Fig. 10 and Supplementary Table 1. The observed congruence in catalytic performance trends suggests that the catalytic performance of 5.9%-HEA (111) is not limited to a particular active center.

Moreover, the reaction processes of HER on 5.9%-HEA (111) without H* spillover are also considered, as shown in Supplementary Fig. 11. For V-H mechanism, the reaction free energy values of PLS are 0.075 eV and 0.099 eV on the weak and strong adsorption sites, respectively. It means that the overpotential values are 0.075/0.099 V on weak/strong adsorption sites, which are much smaller than that (0.375 V) of Pt (111). With the consideration of H* spillover, all the electrochemical steps

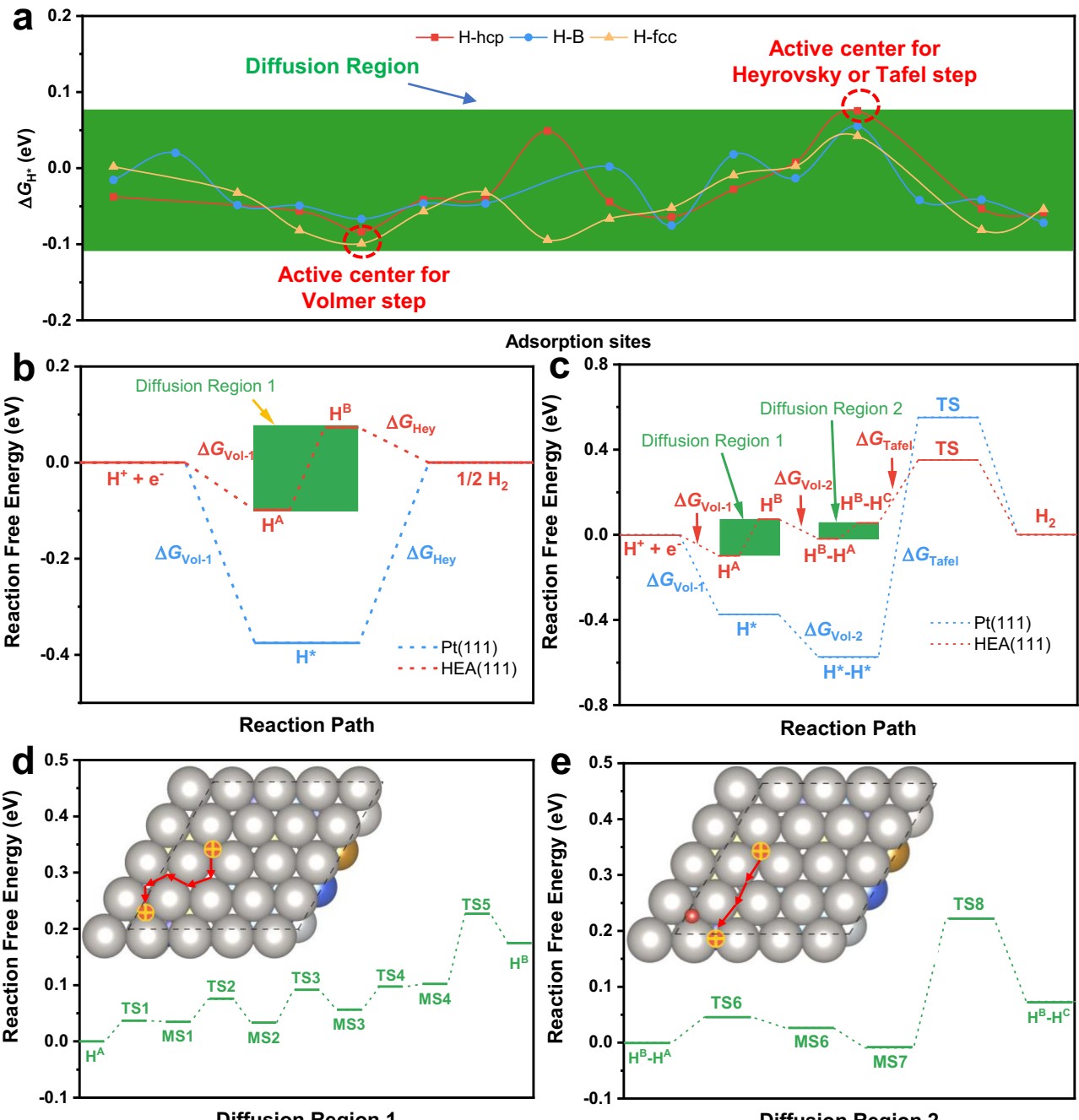

**Fig. 2 | Reaction mechanism studies by DFT calculations. a** The distribution of $\Delta G_{H^*}$ on HEA with 5.9% compressive strain (5.9%-HEA), including different adsorption sites (hcp hollow site, fcc hollow site, and bridge site). The red dashed circles indicate the active centers for Volmer and Heyrovsky (or Tafel) steps, respectively. The green area represents the diffusion region (DR) for the adsorbed H*. **b** Volmer−Heyrovsky mechanism of HER on 5.9%-HEA (111) and Pt (111). **c** Volmer-Tafel mechanism of HER on 5.9%-HEA (111) and Pt (111). **d** The H* spillover on DR1 (diffusion region for the first H*) for 5.9%-HEA (111). **e** The H* spillover on DR2 (diffusion region for the second H*) for 5.9%-HEA (111).

become spontaneous reactions, and the RDS manifests as the H* spillover on the surface of 5.9%-HEA (111) with the largest energy barrier of 0.124 eV. Similarly, for V-T mechanism, the reaction free energy value of the PLS is 0.075 eV on the weak adsorption site, which will become spontaneous electrochemical reaction with considering H* spillover. For the case on the strong adsorption site, the energy barrier is 0.519 eV for the formation of $H_2$, which is larger than that (0.297 eV) with H* spillover. Moreover, the H* spillover is much easier on our designed HEA systems with the smaller energy barrier of ~0.23 eV than those (0.45~1.19 eV) on other reported systems where H* spillover

proved to occur[20,32,33]. It means that the H* spillover should further improve the catalytic activity of HEA systems. As expected, the designed catalysts should have greatly enhanced catalytic performance than Pt. Note that the adsorption sites considered in DFT calculations are very limited relative to those on the HEA surface. In practice, the active centers for the Volmer (Tafel/Heyrovsky) steps should have stronger (weaker) H* adsorption, respectively, which indicates better catalytic activities of HEAs.

Undoubtedly, the influence of strain effect holds a pivotal significance in the enhancement of catalytic performance within HEA

catalysts. Furthermore, our investigation extended to encompass the catalytic process of HER on both the Pt (111) and the unstrained HEA (111) (see Supplementary Fig. 12). Notably, our comparative analysis reveals that, without any strains, HEA catalyst exhibits superior catalytic performance when juxtaposed with Pt (111). This discernible disparity can potentially be attributed to the intricate interplay of adsorption energies of H* stemming from the variegated electronic gradients present across the surface of HEA catalysts. Furthermore, we constructed an idealized 5.9%-PtNi structure comprising one top layer of Pt and four bottom layers of Ni, resulting in a $\Delta G_{H^*}$ of 0.025 eV. This value is comparable to the $\Delta G_{H^*}$ (−0.099–0.075 eV) on 5.9%-HEA (111). The corresponding PLS is the Volmer step with a $\Delta G_{Vol}$ of 0.025 eV and the energy barrier for $H_2$ formation is 0.612 eV, which is still higher than that (0.297 eV) on 5.9%-HEA (111), as illustrated in Supplementary Fig. 13. Such a DFT result suggests that, in addition to the weakened adsorption energy (due to the compressive strain and ligand effects), H* spillover plays a pivotal role in enhancing the catalytic activity. This has also been demonstrated in recent open literatures[20,22,23,34].

## Synthesis and characterization of PtFeCoNiCu HEA

As a proof-of-concept, the PtFeCoNiCu HEA catalysts were synthesized through a solvothermal reaction followed by thermal annealing, as illustrated in Supplementary Fig. 14. Based on different annealing temperatures (300, 400, and 500 °C), the synthesized HEA samples are named as HEA-300, HEA-400, and HEA-500, respectively. The size of the synthesized HEA nanoparticles increases with the annealing temperature, as shown in Supplementary Fig. 15. Figure 3a shows the XRD patterns, where all HEAs present a face-centered cubic structure with three main characteristic peaks corresponding to (111), (200), and (220) planes. The sharp peaks at 15.7° and 16.2° in HEA-300 can be assigned to the transition metal chloride of the precursor, suggesting that 300 °C is not high enough to transform the precursor into HEA thoroughly. Compared with the (111) peak position of Pt/C, the HEA-300, HEA-400, and HEA-500 samples show positive shifts of 1.8°, 2.3°, and 2.3°, respectively, implying the existence of compressive strain in the HEAs caused by alloying with Fe, Co, Ni, and Cu[35,36]. Larger compressive strains appear in HEA-400 and HEA-500 than that in HEA-300, demonstrating the influence of annealing temperature on the strain, which results in the regulation of $\mu$ value in the Gaussian distribution of $\Delta G_{H^*}$. Correspondingly, HEA-300, HEA-400, and HEA-500 present (111) lattice spacings of 0.213, 0.210, and 0.207 nm, respectively, as shown in Supplementary Fig. 16. Compared with the Pt (111) plane (0.226 nm, see Supplementary Fig. 17), HEA-300, HEA-400 and HEA-500 show compressive stains of 5.8%, 7.1% and 8.4%, respectively. It is caused by the alloying of Fe, Co, Ni, and Cu with smaller radii into Pt, which is in accord with the DFT model. Based on the DFT results, the PtFeCoNiCu with a compressive strain of 6.8% should have a $\mu$ value closer to 0 eV, indicating a higher catalytic activity. In agreement with the DFT result, the experimental results indicate HEA-400 with 7.1% strain has a better catalytic performance than HEA-300 (5.8% strain) and HEA-500 (8.4% strain), as shown in Supplementary Fig. 18. Therefore, the system of HEA-400 is our main research object in the following study.

X-ray photoelectron spectroscopy (XPS) analysis (see Supplementary Figs. 19–22) was performed to explore the charge redistribution in HEA catalysts. As shown in Fig. 3b, both $Pt^0$ $4f_{7/2}$ and $4f_{5/2}$ peaks in HEA-400 shift negatively compared with that of Pt/C, demonstrating the electron transfer from other components to Pt in HEA, which agrees well with their electronegativity differences (Fe: 1.83, Co: 1.88, Ni: 1.91, Cu: 1.90, and Pt: 2.28)[30,36,37]. After 5000 cycles of cyclic voltammetry (CV) activating, both the $Pt^0$ $4f_{7/2}$ and $4f_{5/2}$ peaks in HEA-400-5000 (HEA-400 after 5000-cycle CV activation) shift positively compared with those of HEA-400. This is caused by the corrosion of Fe, Co, Ni, and Cu during the activation process. Therefore, reduced electrons are transferred to the Pt element. The transferred electron amounts between Pt atom and its different neighbors are

distinct, leading to a gradient in electron distribution on surface Pt, which is consistent with our proposed DFT model.

Taking HEA-400 as an example, Pt, Fe, Co, Ni, and Cu elements follow an atomic ratio of 18.8: 19.9: 22.3: 20.2: 18.8, according to inductively coupled plasma optical emission spectroscopy (ICP-OES). To exclude the impact of impure phase and to bring component gradient in HEA, CV measurements are conducted for activation at a range of 100–530 mV (vs. RHE). The contents of Fe, Co, Ni and Cu in the electrolyte are detected to increase continuously with the CV activation process (see Supplementary Table 2). To investigate the surface state of HEA-400 after different cycles of CV test, HRTEM characterizations are performed and analyzed. 2000 cycles of CV test are insufficient to introduce component gradient in HEA-400, as shown in Supplementary Fig. 23. After 5000 cycles of CV test, a component gradient appears in HEA-400-5000 (see Supplementary Fig. 24 and Supplementary Note 2). After 10000 cycles of CV test, small Pt particles appear in HEA-400-10000 (see Supplementary Fig. 25). This result indicates that extra CV tests will lead to excessive etching of non-noble elements in HEA-surface, which will reduce the component gradient. All these findings confirm the corrosion of non-Pt elements during the activation. Correspondingly, the average HEA-400 (111) lattice spacings increase with the activation cycles, which states the increase of Pt component proportion (see Supplementary Fig. 26). To further analyze the phase of HEA-400 after different cycles of CV test, XRD characterizations are performed, as shown in Supplementary Fig. 27a. HEA-400-initial presents some impurity peaks corresponding to Co or CoFe species. After 100 cycles of CV test, all the peaks of Co or CoFe species disappear, indicating that HEA-400-100 is a well-defined HEA. The Co and CoFe species that fail to form HEA are etched first during the CV process. In addition to HEA-400-100, HEA-400-2000, HEA-400-5000 and HEA-400-10000 all present well-defined HEA peaks.

A Pt concentration gradient is generated on the surface of the HEA catalyst, as shown in the high-angle annular dark-field scanning TEM (HAADF-STEM) image of HEA-400-5000 (see Fig. 3c). The (111) plane in HEA-surface has a lattice spacing of 0.226 nm (see Supplementary Fig. 28), which is consistent with the Pt (111) spacing, indicating the predominance of Pt in surface layers. The (111) lattice spacing in the core is measured to be 0.210 nm, which is the same as that in HEA-400 before activation (see Supplementary Fig. 16b). Namely, the etching effect only appears in the outermost layers (15–20 layers) of HEA. A (111) lattice spacing in the transition layer is measured to be 0.217 nm, being between those in Pt (111) and HEA-400 (111), indicating the partial etching of non-Pt elements. The gradually diminished lattice spacings from the outermost layers to the core indicate a Pt concentration gradient (see Fig. 3d). The HAADF-EDS elemental maps demonstrate the homogeneously distributed elements in the nanoparticle (see Fig. 3e). Moreover, the Pt concentration gradient is further confirmed in the near-surface elemental maps (see Supplementary Figs. 29–31), where Fe, Co, and Ni are less detected than Cu and Pt, due to the susceptibility to corrosion of Fe, Co, and Ni during the activation process. Furthermore, the HAADF line scan results (see Supplementary Fig. 32 and 33) also display the gradually increased degree of the etching of Fe, Co, and Ni from the core to the outermost layers in HEA-400-5000. All these results indicate that the synthesized HEA catalysts are consistent with the designed dual gradient PtFeCoNiCu HEA model.

## Catalytic performance of PtFeCoNiCu HEA

The HER performance of the HEA catalysts was measured in 0.5 M $H_2SO_4$ using a typical three-electrode configuration. Figure 4a shows polarization curves of HEA-400 before and after CV activation tests. The HEA-400 shows an overpotential of 96.8 mV at −100 mA cm$^{-2}$ in the initial polarization curve. After the 100-cycle CV test, the HEA-400-100 shows a greatly improved HER performance with an overpotential of 37.8 mV at −100 mA cm$^{-2}$. The HER performance of HEA-400 is

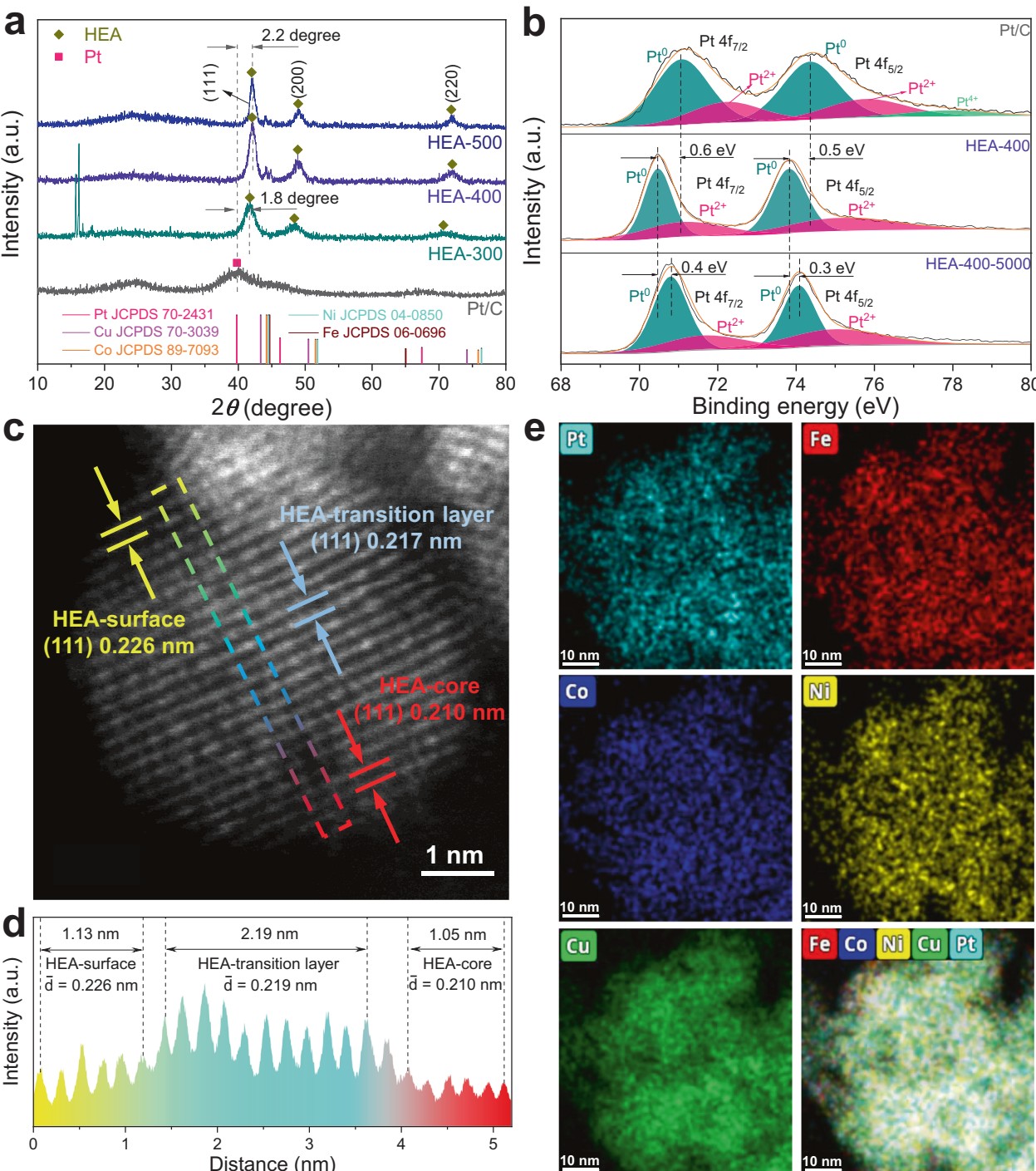

**Fig. 3 | Structural characterization of HEA catalysts. a** XRD patterns of HEA-300, HEA-400, and HEA-500. **b** Pt 4f high-resolution XPS spectra of Pt/C, HEA-400, and HEA-400-5000. **c** HAADF-STEM image of HEA-400-5000. **d** Intensity line profile from the surface to the core in HEA-400-5000 as framed in **c**. **e** HAADF-EDS elemental maps of HEA-400-5000.

gradually enhanced with continuous CV tests until the best performance (the overpotential of 30.7 mV at −100 mA cm⁻²) is achieved after 5000 CV cycles. This is caused by the exposure of more Pt elements on the surface. The Pt, Fe, Co, Ni and Cu elements in HEA-400-5000 follow an atomic ratio of 35.0: 12.3: 14.8: 17.3: 20.6 according to ICP-OES results, consistent with our assumption of the dissolution of Fe, Co, Ni, and Cu components on the surface during the CV activation. Further cycling will make excess Fe, Co, Ni, and Cu elements dissolved, resulting in a smaller electron gradient on the HEA surface and thus a smaller $\sigma$ value. This is the reason that HEA-400-10000 shows an

attenuation performance compared with that of HEA-400-5000 (see Supplementary Fig. 34). Even so, the catalytic activity of HEA-400-10000 is still better than that of Pt/C. Besides, the high stability of HEA-400-5000 is verified through the galvanostatic measurement during an 80-h test (see Supplementary Fig. 35). The faradic efficiency of HEA-400-500 is also calculated to be 99% (see Supplementary Fig. 36), demonstrating that almost all of the current in the test is used for HER. The activation effect of HEA-400 is further demonstrated by the electrochemical double-layer capacitance ($C_{dl}$), as shown in Fig. 4b. Consistent with the tendency of HER activity, the $C_{dl}$ of HEA-400

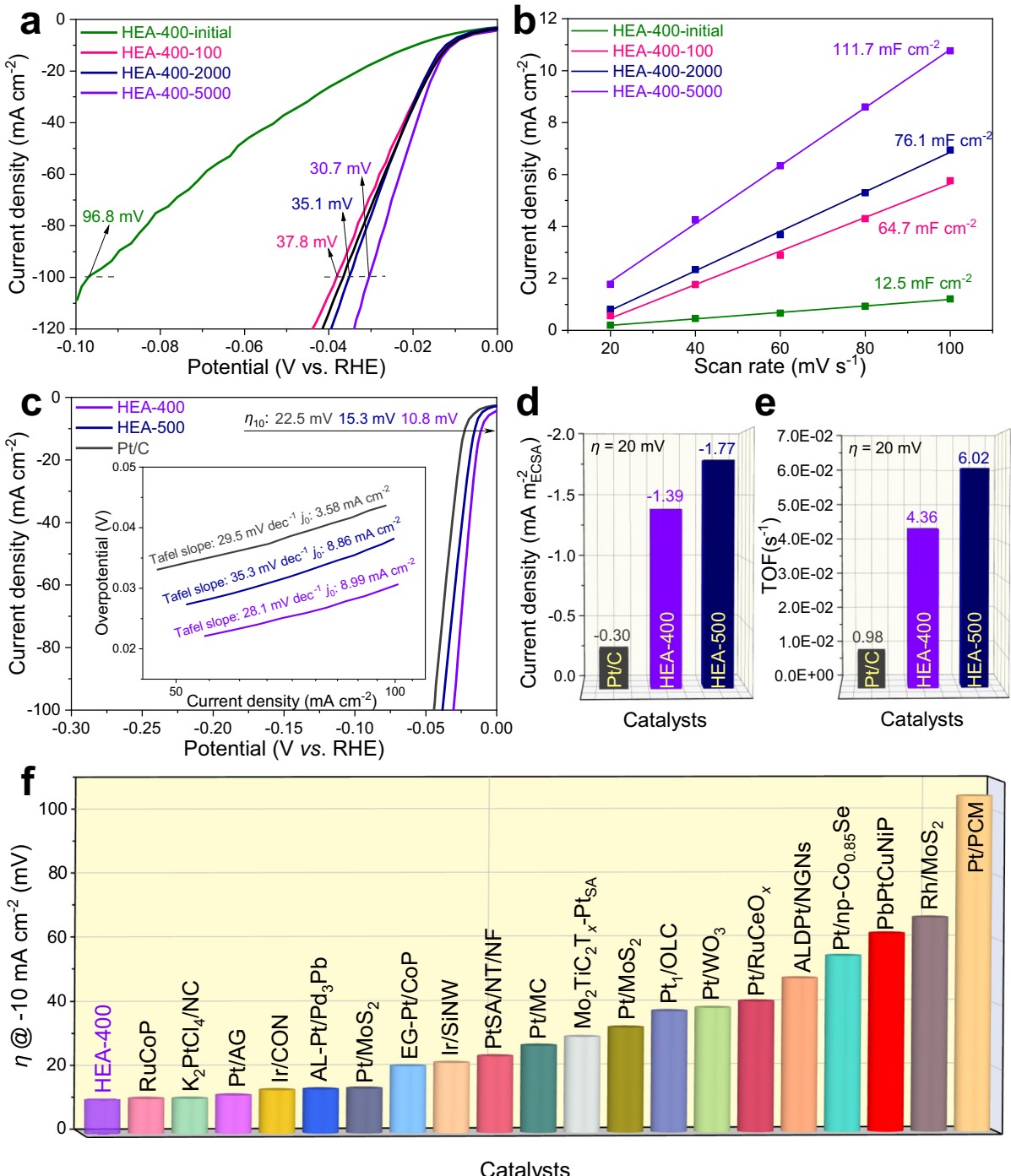

**Fig. 4 | Electrochemical performance. a** Polarization curves of HEA-400 before and after cyclic voltammetry (CV) activation. **b** Plots of capacitive currents with various scan rates for HEA-400 before and after CV activation. **c** Polarization curves of activated HEA-400, activated HEA-500, and Pt/C. The corresponding Tafel plots and exchange current density ($j_0$) are described in the inset. **d** Specific activities of activated HEA-400, activated HEA-500, and Pt/C at $\eta = 20$ mV. **e** TOFs of activated HEA-400, activated HEA-500, and Pt/C at $\eta = 20$ mV. **f** Comparison of HER performance for HEA-400 with other reported catalysts in 0.5 M $H_2SO_4$ (see Supplementary Table 3 for more details).

increases continuously to reach the maximum value of 111.7 mF cm$^{-2}$ after 5000 CV cycles. This indicates the greatly increased electrochemical surface area (ECSA) brought from the dual gradient formed in HEA. More detailed HER performance of HEA-400 before and after CV tests are provided in Supplementary Figs. 37 and 38. In order to highlight the advantages of PtFeCoNiCu HEA catalysts, the HER performances of a ternary PtFeCo alloy and a quaternary PtFeCoNi alloy are compared (see Supplementary Figs. 39–41). As shown in Supplementary Fig. 41, the PtFeCoNiCu catalyst presents better HER

performance than those of PtFeCo and PtFeCoNi, demonstrating the benefits of the dual gradient catalytic system in HEA. In addition, the HER performance of HEA systems without noble metals are measured. As shown in Supplementary Fig. 42, no activation behavior is present in FeCoNiCuMn or FeCoNiCrMn, indicating that the electrochemical activation behavior may exist only in a HEA system with specific active sites, which possess excellent acid corrosion resistance.

The electrochemical activation effect is also observed in HEA-500, as shown in Supplementary Figs. 43 and 44. A comparison of HER

activity among the activated HEA-400 (HEA-400-5000), activated HEA-500 (HEA-500-2000), and Pt/C is shown in Fig. 4c, where the HEA-400 has the smallest overpotential of 10.8 mV at $-10$ mA cm$^{-2}$. The corresponding Tafel plots (see the inset in Fig. 4c) indicate that the activated HEA-400 presents the smallest Tafel slope of 28.1 mV dec$^{-1}$, denoting the best HER kinetics, which is further verified by the largest exchange current density ($j_0 = 8.99$ mA cm$^{-2}$)[38,39]. Additionally, when normalized to ECSA (see Supplementary Fig. 45), HEA-400 presents 4.6 times higher specific electrochemical activity ($-1.39$ mA m$_{ECSA}^{-2}$) than that of Pt/C ($-0.30$ mA m$_{ECSA}^{-2}$, see Fig. 4d) at an overpotential of 20 mV. Meanwhile, the turnover frequency of HEA-400 ($4.36 \times 10^{-2}$ s$^{-1}$) at this potential is also much higher than that of Pt/C ($0.98 \times 10^{-2}$ s$^{-1}$), further confirming the excellent intrinsic activity (see Fig. 4e). The greatly enhanced intrinsic activity of HEA-400 derives from the dual gradient structures, where H* spillover phenomenon exists. The H* spillover phenomenon can be intuitively observed through the color change in the mixture of catalyst and WO$_3$ during the HER process. As shown in Supplementary Fig. 46, the colors of WO$_3$ and Pt/C@WO$_3$ remain unchanged after an 1800-s HER test (the HER processes are presented in Supplementary Fig. 47). However, the color of HEA-400-5000@WO$_3$ changed from dark yellow to dark blue after only a 170-s HER test. This is because the spilled-over hydrogen migrates on HEA surface and readily reacts with WO$_3$ to form dark blue H$_x$WO$_3$[23,40]. The above color change confirms the H* spillover phenomenon on HEA, while no H* spillover occurs on Pt/C. Besides, the H* spillover phenomenon on HEA is verified through the pH-dependent relation of HER. Based on previous reports, the H* spillover based HER catalyst possesses a theoretical reaction order of 2.0 between log |$j$| and pH value[20,41,42]. At a potential of $-0.05$ V (vs. RHE, see Supplementary Fig. 48a), the pH-dependent reaction order for HEA-400-5000 is measured to be 2.03 (see Supplementary Fig. 48b), which is in accord with the theoretical value of 2.0, thus confirming the H* spillover phenomenon. In addition, in situ hydrogen desorption kinetics is studied to further support the H* spillover phenomenon on HEA. To determine the hydrogen desorption kinetics, CV tests are performed from 0.1-0.6 V (vs. RHE) to observe the hydrogen desorption peaks[20]. As shown in Supplementary Fig. 49a, b, the hydrogen desorption peaks shift with increasing scan rates for both Pt/C and HEA-400-5000. Therefore, their hydrogen desorption kinetics can be quantified through comparing the slopes between the hydrogen desorption peak position vs. scan rates. As shown in Supplementary Fig. 49c, HEA-400-5000 presents a significantly reduced slope than Pt/C, demonstrating the improved hydrogen desorption kinetics. Previous study reported that the H* spillover effect can effectively accelerate the hydrogen desorption kinetics for electrocatalysts[20,23,43]. Thus, the accelerated hydrogen desorption kinetics for HEA-400-5000 could derive from the H* spillover phenomenon on the surface. Additionally, the HER performance of HEA-400-5000 was also compared with a PtNi$_3$ system without H* spillover effect, as shown in Supplementary Fig. 50. HEA-400-5000 shows better HER performance than PtNi$_3$ (31.2 mV at $-10$ mA cm$^{-2}$), even though the $\Delta G_{H^*}$ value of PtNi$_3$ may be in close proximity to that of HEA-400-5000 ($\Delta G_{H^*}$ in range around $-0.099$-0.075 eV), indicating the spillover of H* plays a pivotal role in enhancing the catalytic activity. More detailed electrochemical HER comparison between HEA and Pt/C are provided in Supplementary Figs. 51-53. Compared with other reported catalysts for HER (see Fig. 4e and Supplementary Table 3), the designed dual gradient HEA shows the smallest overpotential of 10.8 mV at $-10$ mA cm$^{-2}$, indicating a breakthrough in the catalytic performance for HER.

### Extension to C*, O*, and N*
The unusual Sabatier principle is also extended to other adsorbates (C*, O*, and N*) on the designed HEA catalysts. Based on the BEP relation, the strong adsorption leads to a large diffusion barrier for adsorbates. This means that C*, O*, and N* spillover should be difficult due to their stronger adsorptions than that of H*. Herein, the adsorption energies of C*, O*, and N* on the designed 5.9%-HEA were calculated, as shown in Supplementary Figs. 54-56. Their adsorption energy values also roughly follow the Gaussian distribution with $\mu$ and $\sigma$ values of 1.662 and 0.125 eV, 2.072 and 0.135 eV, 2.103 and 0.104 eV, for C*, O*, and N*, respectively. Similar to the adsorption of H*, the designed 5.9%-HEA still has two active centers with strong and weak adsorptions of these adsorbates. The C*, O*, and N* spillovers between the two active centers are shown in Supplementary Figs. 57-59. The diffusion barrier values of their RDS are 0.772, 0.463, and 0.801 eV for C*, O*, and N*, respectively. Although these diffusion barrier values are larger than that of H*, they are still relatively small values, being sufficient for the spillovers of C*, O*, and N* to occur at room temperature[31]. Therefore, the proposed unusual Sabatier principle in this work can be extended to other catalytic reactions associated with these adsorbates, such as CO$_2$RR, ORR/OER, NRR, etc.

## Discussion
In summary, we discovered the unusual Sabatier principle on HEA catalysts for HER. The new descriptors of $\mu$ and $\sigma$ are proposed to indicate the catalytic activity of HEA catalysts for HER based on the Gaussian distribution [$X \sim N(\mu, \sigma^2)$] of $\Delta G_{H^*}$. Mathematically, a larger $\sigma$ value results with $\mu = 0$ eV results in a higher catalytic activity for HER. Physically, the HEA catalysts with the wider adsorption energy range around 0 eV promoted the adsorption of H* and the formation of H$_2$, simultaneously. According to the new theoretical insights, a PtFeCoNiCu catalyst with dual-gradient (electronic and composition gradients) is precisely synthesized. The designed HEA catalyst has an excellent catalytic performance for HER with an ovepotential of 10.8 mV at $-10$ mA cm$^{-2}$ and 4.6 times higher intrinsic activity than that of Pt/C. The unusual Sabatier principle proved to be applicable to other adsorbents as well. These findings could accelerate the development of HEA catalysts and open avenues to achieve a breakthrough in catalytic performance.

## Methods
### Chemicals
Iron(III) chloride (FeCl$_3$), Chromium(II) chloride (CrCl$_2$), Cobalt(II) chloride hexahydrate (CoCl$_2$·6H$_2$O), Nickel(II) chloride hexahydrate (NiCl$_2$·6H$_2$O), Copper(II) chloride dihydrate (CuCl$_2$·2H$_2$O), Manganese(II) chloride tetrahydrate (MnCl$_2$·4H$_2$O), Chloroplatinic acid (H$_2$PtCl$_6$·6H$_2$O) and polyvinyl pyrrolidone (PVP) were purchased from Sigma Aldrich. All chemicals are of analytical purity and used without further purification.

### Synthesis of PtFeCoNiCu HEA catalyst
0.5 mmol of FeCl$_3$, CoCl$_2$·6H$_2$O, NiCl$_2$·6H$_2$O, CuCl$_2$·2H$_2$O and H$_2$PtCl$_6$·6H$_2$O were added into 40 mL ultrapure water to form a uniform mixture. The mixture was stirred constantly in an oil bath at 80 °C until all water evaporated, forming a dark yellow slurry. The HEA-300, HEA-400 HEA-500, HEA-700 and HEA-900 were obtained through annealing the slurry under 5% H$_2$/Ar atmosphere for 2 h at 300, 400, 500, 700 and 900 °C, respectively.

### Synthesis of PtFeCoNi and PtFeCo catalysts
The synthesis method of PtFeCoNi is the same with that of HEA-400, except that the precursor of CuCl$_2$·2H$_2$O was not added. The synthesis method of PtFeCo is the same with that of HEA-400, except that the precursors of CuCl$_2$·2H$_2$O and NiCl$_2$·6H$_2$O were not added.

### Synthesis of FeCoNiCrMn HEA catalyst
0.5 mmol of FeCl$_3$, CoCl$_2$·6H$_2$O, NiCl$_2$·6H$_2$O, CrCl$_2$ and MnCl$_2$·4H$_2$O were added into 40 mL ultrapure water to form a uniform mixture. The mixture was stirred constantly in an oil bath at 80 °C until all water evaporated, forming a dark gray slurry. The FeCoNiCrMn HEA was

obtained through annealing the slurry under 5% $H_2$/Ar atmosphere for 2 h at 500 °C.

## Synthesis of FeCoNiCuMn HEA catalyst

0.5 mmol of $FeCl_3$, $CoCl_2 \cdot 6H_2O$, $NiCl_2 \cdot 6H_2O$, $CuCl_2 \cdot 2H_2O$ and $MnCl_2 \cdot 4H_2O$ were added into 40 mL ultrapure water to form a uniform mixture. The mixture was stirred constantly in an oil bath at 80 °C until all water evaporated, forming a dark yellow slurry. The FeCoNiCuMn HEA was obtained through annealing the slurry under 5% $H_2$/Ar atmosphere for 2 h at 500 °C.

## Synthesis of PtNi₃ catalyst

The synthesis of $PtNi_3$ was using the method according to the report of Wang et al.[44]. Typically, 0.4 mmol of $H_2PtCl_6 \cdot 6H_2O$, 1.2 mmol $Ni(NO_3)_2 \cdot 6H_2O$ and 287.2 mg PVP were dissolved into 50 mL ultrapure water and sonicated for 1 h. Then, the solution was sprayed onto a glass plate maintained at 400 °C for rapid evaporation. The collected powder was then cleaned through centrifugation for 3 times with ultrapure water. The $PtNi_3$ was obtained through annealing the powder under 5% $H_2$/Ar atmosphere for 2 h at 500 °C.

## Material characterization

For the structural characterization, X-ray diffraction (XRD) was performed on a D/max2500pc diffractometer with a Cu Kα radiation. X-ray photoelectron spectroscopy (XPS) detection was through an ESCALAB 250Xi spectrometer with a monochromatic Al-K source. The morphology characterization was conducted using a JEM-2100F transmission electron microscope (TEM) for TEM images, high-resolution TEM (HRTEM) images and selected area electron diffraction (SAED) patterns. The high angle annular dark field (HAADF) images were obtained through a double-corrected FEI Titan Themis 300 electron microscope. The component analysis was confirmed using an inductively coupled plasma optical emission spectroscopy (ICP-OES).

## Electrochemical measurements

All electrochemical measurements were performed on an Ivium-n-Stat electrochemical workstation under a standard three-electrode system. A graphite electrode, a saturated calomel electrode (SCE) and a rotating disk electrode (RDE) covered with catalyst films were used as the counter electrode, reference electrode, and working electrode, respectively. 3 mg of each catalyst powders distributed into 0.5 mL water-isopropanol solution (4:1, v/v) were used as the catalyst ink. 30 μL of the catalyst ink was taken and dried on the RDE surface for measurement each time. 15 μL Nafion-isopropanol solution (1:19, v/v) was taken and covered on the dried catalyst films as a binder.

The HER performance measurements were conducted in $N_2$-saturated 0.5 M $H_2SO_4$. For the activation of PtFeCoNiCu, PtFeCoNi, PtFeCo, FeCoNiCrMn and FeCoNiCuMn catalysts, cyclic voltammetry (CV) tests were performed at a potential range of 100-530 mV (vs. reversible hydrogen electrode, RHE) at a scan rate of 400 mV s⁻¹. Linear sweep voltammetry (LSV) tests were conducted at a scan rate of 5 mV s⁻¹ with a rotation rate of 2025 rpm. For measuring the double-layer capacitance, CV tests were performed at a potential range of 100 to 200 mV (vs. RHE) at scan rates of 20, 40, 60, 80 and 100 mV s⁻¹, respectively. Galvanostatic tests were performed at an applied current density of −10 mA cm⁻² for 80 h. Electrochemical impedance measurements were performed at 10 mV (vs. RHE) from 100 kHz to 0.1 Hz. The hydrogen production was measured with a gas chromatograph (GC-2014), using a thermal conductivity detector (TCD) to detect $H_2$ content every 10 min. For detecting the amount of produced $H_2$, 1 mg catalyst was loaded on the Ni foam with sufficient Nafion as the binder. All the measurements were performed at room temperature. All the potentials converted to the RHE were through $E_{RHE} = E_{SCE} + 0.059 \text{ pH} + 0.267 \text{ V}$, where $E_{RHE}$ and $E_{SCE}$ denote the

reversible hydrogen evolution potential and the measured potential, respectively.

## Density functional theory calculation

All calculations were performed using the Vienna ab initio simulation package (VASP) based on spin-polarized density function theory (DFT)[45]. The projector-augmented wave pseudopotential was applied to treat the core electrons[46]. The generalized gradient approximation (GGA) with the Perdew–Burke–Ernzerhof functional (PBE) was adopted in the DFT calculations[47]. The kinetic energy cutoff for the wave-function calculations was set to 550 eV. The Fermi smearing function was applied with a smearing width of 0.1 eV. The $4 \times 4$ supercell with five layers was considered for all systems. The Monkhorst-Pack grid of k-points was $2 \times 2 \times 1$, and a vacuum gap of ~15 Å was used to avoid interactions between the system and its mirror images. The geometric relaxation was stopped when the incremental changes in total energy and forces were smaller than $1 \times 10^{-5}$ eV and 0.05 eV/Å, respectively. The van der Waals interaction was considered through the DFT-D3 method proposed by Grimme[48]. All the transition states were obtained using the climbing image nudged elastic band (CI-NEB) method with a convergence force smaller than 0.05 eV/Å[49].

The adsorption energy ($\Delta E_{X^*}$) of adsorbates (X = H, C, O, N) was calculated by the following equation:

$$\Delta Ex^* = Ex^* - E\text{cat} - Ex \qquad (1)$$

where $E_{X^*}$, $E_{cat}$, and $E_X$ represent the total energy of the catalyst with the adsorbate, isolated catalyst, and the corresponding adsorbate, respectively. Note that the energies of H, C, O, and N refer to the energies of $H_2$, graphene, $H_2O$, and $NH_3$, respectively. The adsorption energy was corrected by considering the zero-point energy and entropy, as shown below:

$$\Delta Gx^* = \Delta Ex^* + \Delta ZPE - T\Delta S \qquad (2)$$

where $\Delta G_{X^*}$, $\Delta ZPE$, and $T\Delta S$ denote the adsorption free energy, zero-point energy change, and entropy change, respectively.

## Reporting summary

Further information on research design is available in the Nature Portfolio Reporting Summary linked to this article.

## Data availability

All data are available within the Article, Supplementary Files, and the repository (https://github.com/chandrasinghuoft/PtFeCoNiCu.git), or available from the corresponding authors on reasonable request.

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

## Acknowledgements

We wish to thank the financial supports from the National Natural Science Foundation of China (No. 52130101), Science and Technology Development Program of Jilin Province, China (No. 20230402058GH), Interdisciplinary Integration and Innovation Project of JLU (No. JLUXKJC2021ZY01), the Fundamental Research Funds for the Central Universities, China, the Nature Sciences and Engineer Research Council of Canada (NSERC), Hart Professorship and the University of Toronto. We thank Prof. Edward Sargent for the helpful discussions. We also acknowledge Digital Research Alliance of Canada for providing computing resources at the SciNet, CalculQuebec, and Westgrid consortia. Z.W.C. dedicates this work to his baby, Enxi, who is expected to join his family on Jan. 1st, 2024. May the pursuit of knowledge and scientific inquiry inspire her journey as she grows and explores the world.

## Author contributions

C.C.Y., C.V.S., and Q.J. conceived the project and oversaw all the research phases. Z.W.C. and J.L. designed the project. Q.J., Z.W.C., and P.O. conceived the theoretical model. C.V.S. and Z.W.C. conducted the theoretical calculations. C.C.Y. and J.L. conducted the experiments. Z.W.C., J.L., and P.O. conducted data collection and analysis. J.E.H., Z.W., L.C., X.Y., and G.C. discussed the results and commented on the manuscript.

## Competing interests

The authors declare no competing interests.
