## [Peer Review File · Nature Communications]

REVIEWER COMMENTS

Reviewer #1 (Remarks to the Author):

The authors have investigated the catalytic activity of PtFeCoNiCu high-entropy alloy (HEA) for the electrochemical hydrogen evolution reaction. H* adsorption energies have been simulated with density functional theory on a surface layer of Pt with a Pt concentration gradient down the slab. This has been confirmed with x-ray and electron microscope techniques to be a valid assumption. The authors detect an increased catalytic activity for the HEA compared to pure Pt, arguing that increased compressive strain has lowered the adsorption strength of H*, causing a lower overpotential. Increased number of cycles of cyclic voltammetry was shown to activate the the catalysts, decreasing the overpotential. The investigation is thorough both in terms of simulations and experimental characterization, and represent an avenue for implementing the idea of compressive strain in other systems (although the novelty of this was presented by some of the authors in a previous publication, DOI: 10.1007/s40843-020-1635-9)

My major concern with the current study is the authors' conviction that their increased catalytic activity can be explained by surface diffusion (or "H* spillover").

The proposed "H* spillover" model is enticing, but can be dismissed with fundamental principles unless proof can be given that these fundamental principles are not obeyed.

Let us consider first the Tafel-step of the H-H coupling. According to transition state theory the rate constant of a reaction is proportional to the probability of finding the system at the transition state energy in thermal equilibrium, i.e. $\exp(-\Delta G_{TS} / (k_B T))$. The adsorbates on the surface, however, are moving according to a conservative force, where the path taken by an adsorbate does not influence its potential energy. This means that reaching the transition state energy through thermal fluctuations is independent on whether the adsorbates moved from a strongly adsorbing site to a weakly adsorbing site and then to the transition state (as suggested by e.g. Fig. 2C), or any other path. The rate of the reaction will remain the same, and will uniquely be determined by the energy difference of the most stable state (the thermal ground state) and the transition state.

The same can be argued for the Heyrovsky-step. Here, however, the authors have not considered the transition state (which I agree is reasonable). Assuming, however, the extreme case where the transition state energy is equal to the energy of the product ($\frac{1}{2} H_2$), the rate constant will, as for the Tafel-step, according to transition state theory be uniquely determined by the energy difference

between the strongest adsorption site (the ground state of the system) and, now, the product state. The diffusion is irrelevant according to transition state theory.

The authors will have to provide a justification whether their proposed “anomalous Sabatier principle” with its “diffusion region” is valid in the context of adsorbates moving according to a conservative force. I would argue that the authors’ observation of increased activity on the high-entropy alloy systems could more reasonably be explained by the weakening of the H* adsorption strength by compressive strain, combined with larger than zero coverage of H* at reaction conditions. Both would presumably increase the energy of the ground state of the system relative to pure Pt, leading to faster reaction kinetics without the necessity of “H* spillover”.

Other, but minor, concerns and suggestions are the following:

Line 206: “The transferred electron amounts between Pt atom and its different neighbors are distinct, leading to a gradient in electron distribution on surface Pt, [...]”. Is this evident from Figure 3B? Can the authors elaborate how they observe the “gradient in electron distribution on surface Pt” experimentally?

Line 246: “The HEA-400 shows an overpotential of 96.8 mV at -100 mA cm⁻² in the initial polarization curve”. This potential range is not visible from Figure 4A. The authors should show this to back their claim.

Figure 4B indicates that the electrochemical surface area increases as more cyclic voltammetry cycles are performed. Could the authors elaborate on whether the observed decrease in overpotential in, say, Figure 4A at -100 mA/cm², is solely caused by an increase in surface area or, in fact, also by the dual gradients in their system? The same could be argued for the HEA-500 system in Figure S31A, where HEA-500-2000 appears to have a larger surface area than HEA-500-5000, and also has an increased activity.

Figure 1C: How were the strains obtained to make this plot? By subsurface alloying? In that case what are the compositions and structures used?

Figure 3E: The authors would benefit from larger font size in their element labels as these are barely visible to the reader.

Figure 4C: The font size of the annotated Tafel slopes is a tad too small, and close to being unreadable with the current resolution of the figure.

Figure S27: "Fig. 3B" → "Fig. 4B"

Figure S30: The authors should mention that the alloys studied here are the activated ones, along with their corresponding number of CV cycles.

Figure S33: What is the meaning of the letters "S", "CPE", and "ct"?

Data availability. The conclusions presented by the authors would benefit from transparency in how their analysis was performed by making the raw data publicly available.

Reviewer #2 (Remarks to the Author):

The work is interesting. However, these comments should be answered and considered by the authors.

- 1) As the authors discussed in the paper mainly about the HER of PtFeCoNiCu HEA catalyst, I think the present title is a little broad and may mislead the reader. If the authors think that this can be resolved please consider modification to the title. Also, there is no volcano plot presented in this work and the HEA presented in this work is not anomalous since some bimetallic or ternary metallic catalysts can exhibit this same phenomenon.
- 2) Referring to Figure 6, on this HEA surface, there are surely more than 3 active sites on the surface. Thus, the authors should show and mention if they had considered all these possibilities during the calculation of the adsorption energy.
- 3) There has been no mention of the overpotential of HER in the DFT-calculated model. Please clarify.
- 4) About the simulation part, please provide the height of the vacuum region used in the model to avoid errors in the calculation due to cell periodicity.
- 5) In addition, the K-POINTS used in the calculation is only 2x2x1 which is considered very low to get high accuracy results. Other parameters like the energy convergence of 1×10^{-5} eV are very high

and will give low accuracy. Usually, at least down to 10^{-6} or to 10^{-8} eV would be enough to give high-accuracy results.

6) The force conversion of 0.05 eV/angstrom is also high and will give very low accuracy. At 0.01 eV/angstrom or lower convergence value would be better to guarantee high accuracy result since the H atom is a very small adsorbate.

7) Please provide the detail for the construction of the HEA system used in the DFT calculation. This has to include how the authors design the randomness of the HEA configuration.

8) Referring to the work from Prof. Jens K. Nørskov and the team, they showed that by modifying the sub-atomic later of Pt, the electronics properties, d-band center, and catalytic property can be modified. Please clarify the boon of the work on the enhance catalytic property.

Reviewer #3 (Remarks to the Author):

Based on the PtFeCoNiCu HEA system, this work proposed a “anomalous Sabatier principle” to circumvent the limitations entailed by the well-established Sabatier principle. Specifically, authors state that “Gaussian distribution [$X \sim N(\mu, \sigma^2)$] of ΔGH^* on HEA” could be considered as a descriptor, where a larger σ value results with $\mu = 0$ eV results in a higher catalytic activity for HER. It is indeed a novel idea for explaining the excellent electrocatalytic performance of high-entropy materials derived from the sophisticated “cocktail effect”. However, serious shortcomings and doubtful points exist, which make this paper not ready for being published.

1. No in situ experimental results which can remarkably support the authors’ hypothesis were provided in this work. The authors state that “surface Pt sites with strong H^* adsorption are active centers for the Volmer reaction while the ones with weak H^* adsorption are active centers for the Tafel or Heyrovsky reaction”, but it is known that the sites with strong H^* adsorption are adverse to the desorption of adsorbates and thus how could the diffusion of hydrogen from sites with strong H^* adsorption to the sites with weak H^* adsorption be favorable for the overall reaction?

2. The universality of authors’ viewpoint is limited. First, the HEA is generally a system of solid solution with various metal atoms randomly distributed, which means that the atoms on the surface are arranged in various configurations. As the authors state that “the adsorption sites considered in DFT calculations are very limited relative to those on the HEA surface”, a particular configuration cannot represent the whole system. Second, proposing a untraditional concept with a new descriptor by only one system is obviously not convincing. Whether the HEA system “with a μ value closer to 0 eV and a larger σ value” but without Pt could also reach the similar electrocatalytic activity to PtFeCoNiCu is doubtful.

3. The synthesized materials cannot well match computational model and the proposed concept.

(1) For Fig. 3C, the so-called “HEA-core” region is indeed far from the left edge, but it is also located at the bottom edge, and why this region was marked as “core”. Furthermore, we have carefully checked the interplanar spacings in this image based on the scale bar, and the “HEA-surface”, “HEA-transition layer”, and “HEA-core” were measured to be 0.28, 0.25, and 0.25 nm, obviously different from the values provided by authors (0.226, 0.217, and 0.210 nm).

(2) No obvious composition gradients were observed in Fig. 3E, and the mixed figure of PtFeCoNiCu do not match the mapping images of individual Pt, Fe, Co, Ni, and Cu.

(3) The degree of compressive strain in HEA-300, HEA-400, and HEA-500 were not quantitatively analyzed.

(4) As shown in Fig. 1, authors show that the μ of ΔGH^* could be adjusted by strain, but why the comparison among HEA-300, HEA-400, and HEA-500 was not displayed, especially the HEA-300?

(5) The electrocatalytic activity of PtFeCoNiCu was significantly boosted and individual Pt nanoparticles appeared after the non-noble metals were etched during CV (Fig. S18). Whether the boosted activity is attributed to the newly formed Pt NPs, and whether the PtFeCoNiCu is really outperforms the state-of-the-art catalyst is not clear. If a HEA system without noble metals also display the similar electrochemical behavior? The XRD pattern of the PtFeCoNiCu after CV should be provided.

(6) The HEA with individual Pt NPs not match the computational model in Fig. 1A.

(7) In Fig. 3A, obvious peak at around 43° appear which is very close to the (111) plane of Cu. Further, in Fig. S24 B, the atomic fraction of Cu even exceed 50%. Thus, the synthetic method in this paper could not obtain the well-defined HEA materials.

(8) Whether the individual Cu phase disappear after CV is very suspicious. In Fig. S22-23, the distribution of Cu and Pt is quite similar and authors state that “Fe, Co, and Ni are less detected than Cu and Pt, due to the susceptibility to corrosion of Fe, Co, and Ni during the activation process”. However, in Fig. 24C, the line scan curve of Cu was hidden due to the “inevitable errors” caused by the “equipment factors”, which is quite suspicious.

Reviewer #4 (Remarks to the Author):

In this paper, Chen, Li et al. proposed the anomalous Sabatier principle on high entropy alloys electrocatalysts for hydrogen evolution reaction, where catalytic activity is based on the Gaussian

distribution of ΔG_H^* (influenced by composition, strain, and synthesis conditions). Additionally, those theoretical findings were confirmed by precisely designed catalyst synthesis and experiments.

In my opinion, these findings are of high importance for developing high entropy catalysts.

Therefore, I recommend the publication of this work in Nature Communication after addressing in the revised version the following points:

1. Fig.2. what exactly are DR1 and DR2? Diffusion regions for active sides with strong and weak H bonding?
2. Temperature used by catalyst synthesis is relatively low. Can authors elaborate more about the mixability and stability of this material system (maybe on the examples of phase diagrams)?
3. As the authors also mentioned, catalysts prepared at various temperatures have significantly different electrochemically active surface areas (ECSA). For clarity of presentation and to prove that the observed increase of activity is not only an effect of increased surface area, linear sweep voltammograms in figures: 4A, 4C, S10, and S25 should also be normalized to the ECSA.
4. Potential range of cyclic voltammetry conducted as an activation process should also be mentioned in the main part of the manuscript.
5. How can authors be sure that observed currents are only related to hydrogen evolution reaction (HER)? For example, there might be an overlap of the catalytic current of HER and the current resulting from the dissolution/corrosion of non-noble metal elements. Measuring faradaic efficiencies and/or determining the amount of produced hydrogen might be helpful to answer this question.
6. Caption of figure S21: S20 and S21 should be changed to S22 and S23, respectively.
7. Why was a potential window of 10 mV chosen to determine electrochemically active surface area? It is known from the literature that broader potential windows (100 mV or more) are recommended (see, for example, DOI 10.1088/2515-7655/abee33).
8. Why in comparison plot in Fig. S31 activation of sample HEA-400 and HEA-500 were different?

Detailed Responses to the Comments

Report of Reviewer 1

Q1 (question 1). *The authors have investigated the catalytic activity of PtFeCoNiCu high-entropy alloy (HEA) for the electrochemical hydrogen evolution reaction. H* adsorption energies have been simulated with density functional theory on a surface layer of Pt with a Pt concentration gradient down the slab. This has been confirmed with x-ray and electron microscope techniques to be a valid assumption. The authors detect an increased catalytic activity for the HEA compared to pure Pt, arguing that increased compressive strain has lowered the adsorption strength of H*, causing a lower overpotential. Increased number of cycles of cyclic voltammetry was shown to activate the the catalysts, decreasing the overpotential. The investigation is thorough both in terms of simulations and experimental characterization, and represent an avenue for implementing the idea of compressive strain in other systems (although the novelty of this was presented by some of the authors in a previous publication, DOI: 10.1007/s40843-020-1635-9).*

R1 (reply 1): We sincerely thank the reviewer for reviewing our manuscript and providing positive assessments. As the reviewer said, the novelty of our previous work (DOI: 10.1007/s40843-020-1635-9) is tailoring the lattice strain in high entropy alloys for active and stable methanol oxidation. The compressive strain also plays a vital role in HEA catalysts, but it is not the focus of the current work. In this work, the major novelty is proposing an anomalous Sabatier principle and a new descriptor on HEA catalysts for hydrogen evolution reaction (HER). In addition to the compressive strain contribution, the high catalytic performance of HEA catalysts mainly comes from different active centers on HEA surfaces during HER process and H* spillover between different active centers. Moreover, the ideas presented in this work have the potential to be applied to other reactions. We have revised the manuscript to further highlight the novelty of this work.

Q2. *My major concern with the current study is the authors' conviction that their increased catalytic activity can be explained by surface diffusion (or "H* spillover").*

The proposed "H spillover" model is enticing, but can be dismissed with fundamental principles unless proof can be given that these fundamental principles are not obeyed.*

Let us consider first the Tafel-step of the H-H coupling. According to transition state theory the rate constant of a reaction is proportional to the probability of finding the system at the transition

state energy in thermal equilibrium, i.e. $\exp(-\Delta G_{TS}/(k_B T))$. The adsorbates on the surface, however, are moving according to a conservative force, where the path taken by an adsorbate does not influence its potential energy. This means that reaching the transition state energy through thermal fluctuations is independent on whether the adsorbates moved from a strongly adsorbing site to a weakly adsorbing site and then to the transition state (as suggested by e.g. Fig. 2C), or any other path. The rate of the reaction will remain the same, and will uniquely be determined by the energy difference of the most stable state (the thermal ground state) and the transition state. The same can be argued for the Heyrovsky-step. Here, however, the authors have not considered the transition state (which I agree is reasonable). Assuming, however, the extreme case where the transition state energy is equal to the energy of the product ($\frac{1}{2} H_2$), the rate constant will, as for the Tafel-step, according to transition state theory be uniquely determined by the energy difference between the strongest adsorption site (the ground state of the system) and, now, the product state. The diffusion is irrelevant according to transition state theory.

R2: Thanks for the reviewer's comments. According to transition state theory, the rate constant of "a reaction" is proportional to the probability of finding the system at the transition state energy in thermal equilibrium. Herein, the reaction should be elementary chemical reactions. In practice, a reaction is assumed to be elementary if no reaction intermediates have been detected. Once the active sites of catalysts are complex and diverse, it is likely that there will be multiple reaction intermediates for the adsorption of hydrogen (Nature, 2017, 541, 68-71; Nat. Commun., 2021, 12, 3502; J. Am. Chem. Soc., 2021, 143, 9105-9112). For instance, Dai et al. utilized the H* spillover effect between multifunctional catalytic sites on $La_2Sr_2PtO_{7+x}$ to achieve a high catalytic performance for HER (Nat. Commun., 2022, 13, 1189). Moreover, Jiang et al. demonstrated the H* spillover occurred on Pd₁/Cu (100), where the H spilled from Pd are readily utilized for the semi-hydrogenation of alkynes (Nat. Nanotechnol., 2020, 15, 848-853). Different active centers result in the H* spillover.

In our work, the active sites on HEA surface are more complex. The diffusion of H* is facile with the energy barrier of ~ 0.23 eV, as shown in Fig. 2d-e, which are much smaller than those (0.45 \sim 1.19 eV) in other reported H* spillover mechanism (Nature, 2017, 541, 68-71; Nat. Commun., 2021, 12, 3502; Nat. Commun., 2021, 12, 3884). Therefore, the H* spillover is theoretically feasible on our designed HEA systems.

The H* spillover is further confirmed by experiments in our work. According to previous reports

(Nat. Commun., 2022, 13, 1189; Nat. Commun., 2020, 11, 4773), the H* spillover phenomenon can be intuitively observed through the color change in the mixture of catalyst and WO₃ during the HER process. As shown in Supplementary Fig. 42, the colors of WO₃ and Pt/C@WO₃ remain unchanged after an 1800-s HER test (the HER processes are presented in Supplementary Fig. 43). However, the color of HEA-400-5000@WO₃ changed from dark yellow to dark blue after only a 170-s HER test. This is because the spilled-over hydrogen migrates on HEA surface and readily reacts with WO₃ to form dark blue H_xWO₃ (Nat. Commun., 2022, 13, 1189; J. Phys. Chem., 1964, 68, 411-412). The above color change confirms the H* spillover phenomenon on HEA, while no H* spillover occurs on Pt/C. Besides, the H* spillover phenomenon on HEA is verified through the pH-dependent relation of HER. Based on previous reports (Nat. Commun., 2016, 7, 12272; Adv. Funct. Mater., 2017, 27, 1700359; Nat. Commun., 2021, 12, 3502), the H* spillover based HER catalyst possesses a theoretical reaction order of 2.0 between log |j| and pH value. In our work, at an overpotential of -0.05 V vs. RHE (see Supplementary Fig. 44a), the pH-dependent reaction order for HEA-400-5000 is measured to be 2.03 (see Supplementary Fig. 44b), confirming the H* spillover phenomenon. Finally, in situ hydrogen desorption kinetics is studied to further support the H* spillover phenomenon on HEA. To determine the hydrogen desorption kinetics, CV tests are performed from 0.1 ~ 0.6 V (vs. RHE) to observe the hydrogen desorption peaks (Nat. Commun., 2021, 12, 3502). As shown in Supplementary Fig. 45a-b, the hydrogen desorption peaks shift with increasing scan rates for both Pt/C and HEA-400-5000. Therefore, their hydrogen desorption kinetics can be quantified through comparing the slopes between the hydrogen desorption peak position vs. scan rates. As shown in Supplementary Fig. 45c, HEA-400-5000 presents a significantly reduced slope than Pt/C, demonstrating the improved hydrogen desorption kinetics. Previous study reported that the H* spillover effect can effectively accelerate the hydrogen desorption kinetics for electrocatalysts (Angew. Chem. Int. Ed., 2019, 58, 16038-16042; Nat. Commun., 2021, 12, 3502; Nat. Commun., 2022, 13, 1189). Thus, the accelerated hydrogen desorption kinetics for HEA-400-5000 could derive from the H* spillover phenomenon on the surface.

Based on the above discussions, the H* spillover model divides hydrogen evolution reactions into several elementary chemical reactions through finding some intermediates of H* absorbed on the different adsorption sites of HEA surfaces. The fundamental principle of transition state theory still works for each elementary chemical reaction. Both DFT calculations and experimental results

prove the feasibility of H* spillover. More details are shown in the revised manuscript.

“Moreover, the H* spillover is much easier on our designed HEA systems with the smaller energy barrier of ~ 0.23 eV than those (0.45 \sim 1.19 eV) on other reported systems where H* spillover proved to occur.²⁸⁻³⁰” in lines 11-13 of page 5.

“The H* spillover phenomenon can be intuitively observed through the color change in the mixture of catalyst and WO₃ during the HER process. As shown in Supplementary Fig. 42, the colors of WO₃ and Pt/C@WO₃ remain unchanged after an 1800-s HER test (the HER processes are presented in Supplementary Fig. 43). However, the color of HEA-400-5000@WO₃ changed from dark yellow to dark blue after only a 170-s HER test. This is because the spilled-over hydrogen migrates on HEA surface and readily reacts with WO₃ to form dark blue H_xWO₃.^{36,37} The above color change confirms the H* spillover phenomenon on HEA, while no H* spillover occurs on Pt/C. Besides, the H* spillover phenomenon on HEA is verified through the pH-dependent relation of HER. Based on previous reports, the H* spillover based HER catalyst possesses a theoretical reaction order of 2.0 between $\log |j|$ and pH value.^{29,38,39} At an overpotential of -0.05 V (vs. RHE, see Supplementary Fig. 44a), the pH-dependent reaction order for HEA-400-5000 is measured to be 2.03 (see Supplementary Fig. 44b), which is in accord with the theoretical value of 2.0, thus confirming the H* spillover phenomenon. In addition, *in situ* hydrogen desorption kinetics is studied to further support the H* spillover phenomenon on HEA. To determine the hydrogen desorption kinetics, CV tests are performed from 0.1 \sim 0.6 V (vs. RHE) to observe the hydrogen desorption peaks.²⁹ As shown in Supplementary Fig. 45a-b, the hydrogen desorption peaks shift with increasing scan rates for both Pt/C and HEA-400-5000. Therefore, their hydrogen desorption kinetics can be quantified through comparing the slopes between the hydrogen desorption peak position vs. scan rates. As shown in Supplementary Fig. 45c, HEA-400-5000 presents a significantly reduced slope than Pt/C, demonstrating the improved hydrogen desorption kinetics. Previous study reported that the H* spillover effect can effectively accelerate the hydrogen desorption kinetics for electrocatalysts.^{29,36,40} Thus, the accelerated hydrogen desorption kinetics for HEA-400-5000 could derive from the H* spillover phenomenon on the surface.” in lines 11-34 of page 8.

Fig. 2. (d) The H* spillover on DR1 (diffusion region for the first H*) for 5.9%-HEA (111). (e) The H* spillover on DR2 (diffusion region for the second H*) for 5.9%-HEA (111).

Supplementary Fig. 42. Color change photographs of (a) WO_3 , (b) mixture of Pt/C@WO_3 and (c) mixture of HEA-400-5000@WO_3 . The photographs are taken before and after the HER process as shown in Supplementary Fig. 43.

Supplementary Fig. 43. HER galvanostatic plots for WO_3 , Pt/C@WO_3 and HEA-400-5000@WO_3 to identify the spillover mechanism.

Supplementary Fig. 44. (a) Tafel curves of HEA-400-5000 in H_2SO_4 with pH ranging from 0 to 0.4. (b) The liner plot of $\log |j|$ at -0.05 V (vs. RHE) vs. pH for HEA-400-5000 .

Supplementary Fig. 45. (a) CV profiles of HEA-400-5000 with the scan rate from 65 to 850 mV s⁻¹ in 0.5 M H₂SO₄. (b) CV profiles of Pt/C with the scan rate from 65 to 850 mV s⁻¹ in 0.5 M H₂SO₄. (c) Plots of hydrogen desorption peak position vs. scan rates for HEA-400-5000 and Pt/C.

Q3. *The authors will have to provide a justification whether their proposed “anomalous Sabatier principle” with its “diffusion region” is valid in the context of adsorbates moving according to a conservative force. I would argue that the authors’ observation of increased activity on the high-entropy alloy systems could more reasonably be explained by the weakening of the H* adsorption strength by compressive strain, combined with larger than zero coverage of H* at reaction conditions. Both would presumably increase the energy of the ground state of the system relative to pure Pt, leading to faster reaction kinetics without the necessity of “H* spillover”.*

R3: We agree with the reviewer’s argument, the weakening of the H adsorption by compressive strain on HEA systems would accelerate reaction kinetics, resulting in the increased catalytic activity on HEA catalysts. We have calculated the reaction process on HEA surface without “diffusion region”, and the corresponding results are shown in Supplementary Fig. 9. For Volmer-Heyrovsky mechanism, the reaction free energy values of the potential limiting step are 0.075 eV and 0.099 eV on the weak and strong adsorption sites, respectively, which are much smaller than that of Pt (111) (0.375 eV). When the diffusion region is considered, all the electrochemical steps become spontaneous reactions, indicating better catalytic activity. Similarly, for Volmer-Tafel mechanism, the reaction free energy value of the potential limiting step is 0.075 eV on the weak adsorption site, which will become spontaneous electrochemical reaction when considering “diffusion region”. For the case on the strong adsorption site, the energy barrier for H₂ formation is 0.519 eV, which is larger than that (0.297 eV) with “diffusion region”. This means that the introduction of “diffusion region” would further improve the catalytic activity of HEA systems. More importantly, the H* spillover is further confirmed by DFT calculations and experiments in this work (please see **R2**). More details are shown in lines 1-14 of page 5.

“Moreover, the reaction processes of HER on 5.9%-HEA (111) without H* spillover are also considered, as shown in Supplementary Fig. 9. For V-H mechanism, the reaction free energy values of PLS are 0.075 eV and 0.099 eV on the weak and strong adsorption sites, respectively. It means that the overpotential values are 0.075/0.099 V (vs. RHE) on weak/strong adsorption sites, which are much smaller than that (0.375 V vs. RHE) of Pt (111). When H* spillover is considered, all the electrochemical steps become spontaneous reactions, indicating the better catalytic activity for HER. Similarly, for V-T mechanism, the reaction free energy value of the PLS is 0.075 eV on the weak adsorption site, which will become spontaneous electrochemical reaction with considering H* spillover. For the case on the strong adsorption site, the energy barrier is 0.519 eV for the formation of H₂, which is larger than that (0.297 eV) with H* spillover. Moreover, the H* spillover is much easier on our designed HEA systems with the smaller energy barrier of ~ 0.23 eV than those (0.45 ~ 1.19 eV) on other reported systems where H* spillover proved to occur.²⁸⁻³⁰ It means that the H* spillover should further improve the catalytic activity of HEA systems.”

Supplementary Fig. 9. (a) Volmer-Heyrovsky mechanism of HER on 5.9%-HEA (111) without H* spillover. (b) Volmer-Tafel mechanism of HER on 5.9%-HEA (111) without H* spillover.

Other, but minor, concerns and suggestions are the following:

Q4. Line 206: “The transferred electron amounts between Pt atom and its different neighbors are distinct, leading to a gradient in electron distribution on surface Pt, [...]”. Is this evident from Figure 3B? Can the authors elaborate how they observe the “gradient in electron distribution on surface Pt” experimentally?

R4: Thanks for the reviewer’s comments. Fig. 3b only shows the electron transfer from other components to Pt, and we can’t find the gradient in electron distribution on surface Pt. It is very difficult to directly observe the gradient in electron distribution on surface Pt due to limited experimental conditions. Alternatively, to further demonstrate this point, we have calculated the Bader charge distribution, as shown in Supplementary Fig. 1. Combined with the electron

localization function contour for HEA (111) plane (see Fig. 1b), obviously, the surface Pt atoms accept different numbers of electrons due to the different neighbors. Moreover, the H* spillover has been proved in our experiments (please see **R2** and **R3**). This means that the adsorption of H* is diverse on HEA (111), which could indirectly prove the electron distribution gradient on the HEA (111). More details are shown in lines 12-15 of page 3.

“The coordination atoms to surface Pt active sites are diverse due to the nature of HEA, which results in various electron redistributions (electronic gradient), as shown in Fig. 1b. The corresponding Bader charge distribution is described in Supplementary Fig. 1. Such an electronic gradient causes different adsorption abilities for H*”

Supplementary Fig. 1. The geometrically optimized structure of PtFeCoNiCu (111). The smaller balls indicate the surface Pt atoms, and the number represents the transferred electron amounts from neighbors to the corresponding Pt atoms.

Q5. Line 246: “The HEA-400 shows an overpotential of 96.8 mV at -100 mA cm^{-2} in the initial polarization curve”. This potential range is not visible from Figure 4A. The authors should show this to back their claim.

R5: Following reviewer’s suggestion, polarization curves of HEA-400 before and after cyclic voltammetry with a potential range of $0 \sim -100 \text{ mV}$ (vs. RHE) are provided in the revised manuscript, where the overpotential of 96.8 mV at -100 mA cm^{-2} in HEA-400-initial can be clearly observed (see Fig. 4a).

Fig. 4a. Polarization curves of HEA-400 before and after cyclic voltammetry (CV) activation.

Q6. Figure 4B indicates that the electrochemical surface area increases as more cyclic voltammetry cycles are performed. Could the authors elaborate on whether the observed decrease in overpotential in, say, Figure 4A at -100 mA/cm^2 , is solely caused by an increase in surface area or, in fact, also by the dual gradients in their system? The same could be argued for the HEA-500 system in FigureS31A, where HEA-500-2000 appears to have a larger surface area than HEA-500-5000, and also has an increased activity.

R6: To explore the origin of HER activity for HEA-400 and HEA-500 with different cycles of CV test, the corresponding ECSA and specific activity are compared. The ECSA is determined by the double-layer capacitance (C_{dl}). 100 cycles of CV test are performed before the specific activity comparison to exclude the impact of impure phase or sample surface oxidation. For HEA-400, the C_{dl} increases continuously with the CV progress and reaches the maximum value of 111.7 mF cm^{-2} after 5000 CV cycles (see Supplementary Fig. 33b). The greatly increased C_{dl} derives from the gradually increasing exposed Pt sites and the enlarged component gradient during the CV test. Further CV tests will reduce the C_{dl} of HEA-400 (the C_{dl} of HEA-400-10000 is 95.0 mF cm^{-2}), which may be attributed to the diminished component gradient in the surface after excessive CV tests. Corresponding to C_{dl} , the HER activity of HEA-400 continues to improve with the CV progress and HEA-400-5000 shows the best performance with an η_{100} of 30.7 mV (see Supplementary Fig. 33a). Further CV tests will reduce the HER activity of HEA-400, and the HEA-400-10000 sample shows an increased η_{100} of 36.6 mV. When normalized to the ECSA, the specific activity of HEA-400 declines slightly with the CV progress. This is because the electron gradient in HEA surface gradually decreases during the CV progress. Above all, the improvement in HER

activity of HEA-400 during the CV activation is solely caused by an increase in ECSA.

For HEA-500, the C_{dl} and HER activity increase continuously with the CV progress and reach the best performance after 2000 CV cycles (see Supplementary Fig. 39). The HEA-500-2000 shows a C_{dl} of 40.5 mF cm^{-2} and an η_{100} of 42.1 mV. Further CV tests will reduce the C_{dl} and HER activity of HEA-500. HEA-500-5000 shows a diminished C_{dl} of 34.9 mF cm^{-2} and an increased η_{100} of 62.5 mV. When normalized to the ECSA, the specific activity of HEA-500 declines slightly with the CV progress, which is also attributed to the diminished electron gradient in the surface. Thus, the improvement in HER activity of HEA-500 during the CV activation is also caused by an increase in ECSA.

Although the specific activity for HEA-400 and HEA-500 slightly declines with increasing CV cycles, the activated HEA-400 and HEA-500 still present superior specific activity than Pt/C (see Supplementary Fig. 41). This is because HEA-400-5000 and HEA-500-2000 still possess stronger surface electron gradients than Pt/C. Combined with the component gradient formed during the CV progress, the superior HER activity of the activated HEA than Pt/C derives from the dual gradients catalytic system. The details have been added into the revised supplementary material.

“For HEA-400, the C_{dl} increases continuously with the CV progress and reaches the maximum value of 111.7 mF cm^{-2} after 5000 CV cycles (see Supplementary Fig. 33b). The greatly increased C_{dl} derives from the gradually increasing exposed Pt sites and the enlarged component gradient during the CV test. Further CV tests will reduce the C_{dl} of HEA-400 (the C_{dl} of HEA-400-10000 is 95.0 mF cm^{-2}), which may be attributed to the diminished component gradient in the surface after excessive CV tests. Corresponding to the C_{dl} , the HER activity of HEA-400 continues to improve with the CV progress and HEA-400-5000 shows the best performance with an η_{100} of 30.7 mV (see Supplementary Fig. 33a). Further CV tests will reduce the HER activity of HEA-400, and the HEA-400-10000 sample shows an increased η_{100} of 36.6 mV. When normalized to the ECSA, the specific activity of HEA-400 declines slightly with the CV progress. This is because the electron gradient in HEA surface gradually decreases during the CV progress.” Supplementary Note 6.

“For HEA-500, the C_{dl} and HER activity increase continuously with the CV progress and reach the best performance after 2000 CV cycles. The HEA-500-2000 shows a C_{dl} of 40.5 mF cm^{-2} and an η_{100} of 42.1 mV. Further CV tests will reduce the C_{dl} and HER activity of HEA-500. HEA-500-5000 shows a diminished C_{dl} of 34.9 mF cm^{-2} and an increased η_{100} of 62.5 mV. When normalized to the ECSA, the specific activity of HEA-500 declines slightly with the CV progress, which is also attributed to the diminished electron gradient in the surface.” Supplementary Note 7.

“The activated HEA-400 and HEA-500 present superior specific activity than Pt/C. This is because HEA-400-5000 and HEA-500-2000 still possess stronger surface electron gradients than Pt/C.

Combined with the component gradient formed during the CV progress, the superior HER activity of the activated HEA than Pt/C derives from the dual gradients catalytic system.” Supplementary Note 9.

Supplementary Fig. 33. (a) Polarization curves of HEA-400 with different cycles of CV activation. (b) Plots of capacitive currents with various scan rates for HEA-400 with different cycles of CV activation. (c) Polarization curves normalized to ECSA for HEA-400 with different cycles of CV activation.

Supplementary Fig. 39. (a) Polarization curves of HEA-500 with different cycles of CV activation. (b) Plots of capacitive currents with various scan rates for HEA-500 with different cycles of CV activation. (c) Polarization curves normalized to ECSA for HEA-500 with different cycles of CV activation.

Supplementary Fig. 41. Polarization curves normalized to ECSA of HEA-400-5000, HEA-500-2000, and Pt/C.

Q7. Figure 1C: How were the strains obtained to make this plot? By subsurface alloying? In that

case what are the compositions and structures used?

R7: We apologize for the confusion on the strains. For DFT models, the strains were obtained by changing the lattice constant of HEA systems, which could be achieved by the composition regulation and changing the synthesis condition in experiments (for example, the annealing temperature, *Sci. China Mater.*, 2021, 64, 2454-2466). The details have been added in lines 15-17 of page 4 in Supplementary Methods.

“To simulate the strain effect for the adsorption of H*, the strains were obtained by changing the lattice constant of HEA (111) without considering the composition changes.”

Q8. *Figure 3E: The authors would benefit from larger font size in their element labels as these are barely visible to the reader.*

R8: As suggested, the font size of element labels in Fig. 3e have been enlarged.

Fig. 3e. HAADF-EDS elemental maps of HEA-400-5000.

Q9. *Figure 4C: The font size of the annotated Tafel slopes is a tad too small, and close to being*

unreadable with the current resolution of the figure.

R9: Following reviewer's suggestion, the font size in the Tafel plots in Fig. 4c has been enlarged to make it more readable.

Fig. 4c. Polarization curves of activated HEA-400, activated HEA-500, and Pt/C. The corresponding Tafel plots and exchange current density (j_0) are described in the inset.

Q10. Figure S27: “Fig. 3B” → “Fig. 4B”

R10: We are sorry for the unmatched caption provided in Supplementary Fig. 27 in the original manuscript. The caption of Supplementary Fig. 34 (original Supplementary Fig. 27) has been revised.

Supplementary Fig. 34. (a-e) Detailed cyclic voltammetry data for HEA-400-initial, HEA-400-100, HEA-400-2000, HEA-400-5000 and HEA-400-10000 to determine the double layer capacitance, respectively.

Q11. Figure S30: The authors should mention that the alloys studied here are the activated ones,

along with their corresponding number of CV cycles.

R11: Following reviewer's suggestion, the activation states of the alloys in Supplementary Fig. 37 (original Supplementary Fig. 30) have been claimed in the figure caption.

Supplementary Fig. 37. Comparison of polarization curves among PtFeCo ternary alloy (PtFeCo-initial), PtFeCoNi quaternary alloy (PtFeCoNi-2000) and PtFeCoNiCu HEA (HEA-400-5000).

Q12. Figure S33: What is the meaning of the letters “S”, “CPE”, and “ct”?

R12: R_s , R_{ct} and CPE denote the solution resistance, charge transfer resistance and constant phase element, respectively. The corresponding description has been added into the caption of Supplementary Fig. 46 (original Supplementary Fig. 33).

Supplementary Fig. 46. Nyquist plots of activated HEA-400 (HEA-400-5000), activated HEA-500 (HEA-500-2000), and Pt/C, the inset shows the equivalent electrical circuit diagram, where R_s , R_{ct} and CPE denote the solution resistance, charge transfer resistance and constant phase element, respectively.

Q13. Data availability. The conclusions presented by the authors would benefit from transparency

in how their analysis was performed by making the raw data publicly available.

R13: We strongly agree with the reviewer's view. We have combined the corresponding data into an excel spreadsheet, as Supplementary Data. All data are available within the Article and Supplementary Files, or available from the corresponding authors on reasonable request.

Reviewer #2:

Q. *The work is interesting. However, these comments should be answered and considered by the authors.*

R: Thanks for the reviewer's positive comments. We have revised the manuscript according to the reviewer's comments/suggestions one by one.

Q1. *As the authors discussed in the paper mainly about the HER of PtFeCoNiCu HEA catalyst, I think the present title is a little broad and may mislead the reader. If the authors think that this can be resolved please consider modification to the title. Also, there is no volcano plot presented in this work and the HEA presented in this work is not anomalous since some bimetallic or ternary metallic catalysts can exhibit this same phenomenon.*

R1: We agree with the reviewer that the title is a little broad and there is no volcano plot presented in this work. Therefore, the title has been changed to "Anomalous Sabatier principle on high entropy alloy catalysts for hydrogen evolution reactions". Most bimetallic or ternary metallic catalysts still follow the Sabatier principle (Nat. Mater., 2006, 5, 909-913; Science, 2016, 352, 73-76; Nat. Commun., 2018, 9, 3702). They modulated the adsorption energy value by alloying to make it closer to the optimal value (neither too strong nor too weak: traditional Sabatier principle). In this work, the adsorption energy value on HEA catalysts is a range (Gaussian distribution), rather than a single or several values. The strong adsorption sites are expected to be stronger, which accelerates the Volmer step of HER; while the weak adsorption sites are expected to be weaker, which promotes the Heyrovsky/Tafel step of HER. We define this phenomenon as the anomalous Sabatier principle, which is unique on HEA catalysts for HER. The details have been added in the last line of page 3 and lines 1-6 of page 4.

“Different from the traditional bimetallic and ternary metallic catalysts,^{14,22,26} the active sites of HEA are more diverse and their ΔG_{H^*} values follow a Gaussian distribution, rather than a definite value. Hence, the Sabatier principle and the criterion of $\Delta G_{H^*} = 0$ eV are no longer valid for HEA

catalysts. In this work, we propose an anomalous Sabatier principle, where the Gaussian distribution of ΔG_{H^*} with a μ value closer to 0 eV and a larger σ value on HEA catalysts could be used as the descriptor of the higher catalytic activity for HER.”

Q2. Referring to Figure 6, on this HEA surface, there are surely more than 3 active sites on the surface. Thus, the authors should show and mention if they had considered all these possibilities during the calculation of the adsorption energy.

R2: We are sorry for the confusion on active sites. Supplementary Fig. 7 shows 3 types of active sites on HEA (111). In fact, we have calculated more than 400 active sites on the 4×4 HEA (111). Admittedly, it is almost impossible to consider all possibilities due to the complex active sites on the surface of HEA. For instance, a HEA with a face-centered cubic (FCC) structure containing five elements should have 5^{15} active bridge sites on its (111) surface when only the bridge site (2 atoms) and the nearest neighbor atoms (13 atoms) are considered active centers. Therefore, we selected 48 active sites to calculate ΔG_{H^*} on one HEA surface, which follows a Gaussian distribution. The fitting Gaussian distribution could roughly indicate the adsorption properties of this HEA. The adsorption sites considered in DFT calculations are very limited relative to those on the HEA surface. In practice, the active centers for the Volmer (Tafel/Heyrovsky) steps should have stronger (weaker) H^* adsorption, which indicates better catalytic activities of HEAs. Detailed description is presented in the revised manuscript.

“ ΔG_{H^*} is calculated on the designed HEA (111) with different strains (see Fig. 1d), including more than 400 datapoints. The ΔG_{H^*} distribution roughly conforms to the Gaussian distribution [$X \sim N(\mu, \sigma^2)$]. Herein, μ and σ^2 determine the location and the variance of ΔG_{H^*} , respectively” in lines 31-33 of page 3.

“Note that the adsorption sites considered in DFT calculations are very limited relative to those on the HEA surface. In practice, the active centers for the Volmer (Tafel/Heyrovsky) steps should have stronger (weaker) H^* adsorption, respectively, which indicates better catalytic activities of HEAs.” in lines 15-18 of page 5.

Supplementary Fig. 7. Adsorption site types of the bridge, fcc hollow, and hcp hollow sites for H* on HEA (111).

Q3. *There has been no mention of the overpotential of HER in the DFT-calculated model. Please clarify.*

R3: As suggested, some discussions on the overpotential of HER have been added into the revised manuscript, as shown in lines 2-7 of page 5.

“For V-H mechanism, the reaction free energy values of PLS are 0.075 eV and 0.099 eV on the weak and strong adsorption sites, respectively. It means that the overpotential values are 0.075/0.099 V (vs. RHE) on weak/strong adsorption sites, which are much smaller than that (0.375 V vs. RHE) of Pt (111). When H* spillover is considered, all the electrochemical steps become spontaneous reactions, indicating the better catalytic activity for HER.”

Q4. *About the simulation part, please provide the height of the vacuum region used in the model to avoid errors in the calculation due to cell periodicity.*

R4: Following reviewer’s suggestion, the height of the vacuum region (~15 Å) has been provided in the model, which is large enough to avoid errors in the calculation due to cell periodicity. Detailed description is presented in lines 13-15 of page 11.

“The Monkhorst-Pack grid of k-points was $2 \times 2 \times 1$, and a vacuum gap of ~ 15 Å was used to avoid interactions between the system and its mirror images.”

Q5. *In addition, the K-POINTS used in the calculation is only $2 \times 2 \times 1$ which is considered very low to get high accuracy results. Other parameters like the energy convergence of 1×10^{-5} eV are very high and will give low accuracy. Usually, at least down to 10^{-6} or to 10^{-8} eV would be enough to give high-accuracy results.*

R5: As suggested, the reaction processes of HER on HEA (111) and Pt (111) are recalculated by

using the k-points of $4 \times 4 \times 1$, the energy convergence of 1×10^{-8} eV, and force convergence of 0.01 eV/angstrom, as shown in Supplementary Fig. 55. We achieved the desired high accuracy results. Compared with our previous results, the differences are very small (see Supplementary Table 3), indicating the reliability of our DFT calculations. In this work, we focus on the relative energy values (such as: adsorption energy, reaction free energy, energy barrier), rather than the absolute energy values. Therefore, medium-precision calculation parameters are enough to obtain a convincing result, as similar as the published works (ACS Catal., 2016, 6, 4428-4437; Science, 2016, 352, 6281). The details are shown in lines 5-8 of page 4 in Supplementary Methods.

“To check the accuracy of calculation parameters, higher accuracy parameters were considered to calculate the reaction process of HER on the designed HEA (111), as shown in Supplementary Fig. 55. The relatively small differences (Supplementary Table 3) indicate the reliability of our DFT calculations.”

Supplementary Fig. 55. (A) Volmer-Heyrovsky mechanism and (B) Volmer-Tafel mechanism of HER on 5.9%-HEA (111) and Pt (111) under the high accuracy calculation parameters (k-point: $4 \times 4 \times 1$; energy convergence: 1×10^{-8} eV; force convergence: 0.01 eV/Å)

Supplementary Table 3. Reaction free energy values of all steps during HER on HEA (111) and Pt (111) with low (k-point: $2 \times 2 \times 1$; energy convergence: 1×10^{-5} eV; force convergence: 0.05 eV/Å)/high (k-point: $4 \times 4 \times 1$; energy convergence: 1×10^{-8} eV; force convergence: 0.01 eV/Å) precision calculation parameters.

Reactions	Catalysts	Low Accuracy	High Accuracy
$\Delta G_{\text{Vol-1}}$ (eV)	Pt (111)	-0.375	-0.365
	HEA (111)	-0.099	-0.082
$\Delta G_{\text{Vol-2}}$ (eV)	Pt (111)	-0.201	-0.185
	HEA (111)	-0.091	-0.089
ΔG_{Hey} (eV)	Pt (111)	0.375	0.365
	HEA (111)	0.075	0.077
ΔG_{Tafel} (eV)	Pt (111)	1.128	1.063
	HEA (111)	0.297	0.297

Q6. *The force conversion of 0.05 eV/angstrom is also high and will give very low accuracy. At 0.01 eV/angstrom or lower convergence value would be better to guarantee high accuracy result since the H atom is a very small adsorbate.*

R6: Please see **R5**.

Q7. *Please provide the detail for the construction of the HEA system used in the DFT calculation. This has to include how the authors design the randomness of the HEA configuration.*

R7: Following reviewer's suggestion, we have provided the detail for the construction of the HEA system used in the DFT calculation, as shown in lines 10-16 of page 4 in Supplementary Methods.

“The HEA (111) structures used for DFT calculations were initially created from FCC lattice structures. 4×4 supercell with five layers was selected as the model. The order of the lattice points in each simulation was randomly shuffled and the shuffled lattice points were assigned different types of metal atoms consistent with the corresponding element ratio. The lattice constant of the generated HEA (111) was received by the optimization of geometric structures, and the surface interaction with adsorbates were calculated by relaxing the adsorbates and the top three layers while fixing the bottom two layers.”

Q8. *Referring to the work from Prof. Jens K. Norskov and the team, they showed that by modifying the sub-atomic later of Pt, the electronics properties, d-band center, and catalytic property can be modified. Please clarify the boon of the work on the enhance catalytic property.*

R8: Thanks for the reviewer's comment. Prof. Norskov's group showed that the catalytic performance of Pt could be improved by modifying the sub-atomic layer (Science, 2016, 352, 73-76). In our system, the catalytic performance of Pt is enhanced by the different coordination environments in HEA, which results in the gradient electron distribution on the surface. This is the biggest difference from Prof. Norskov's work. The gradient electron distribution on the surface enables the adsorption of hydrogen to exhibit a Gaussian distribution. The active sites with strong adsorption adsorb hydrogen (H*) and other sites with weak adsorption release H* to produce H₂, giving rise to high catalytic performance for HER. Based on this phenomenon, we proposed the anomalous Sabatier principle with a new descriptor for HER on HEA catalysts, which is the most novel aspect of our work. We have revised this part of the statement to make the point stronger, as shown in the last two lines of page 3 and lines 1-6 of page 4.

“As is well known, $\Delta G_{H^*} = 0$ eV denotes the optimal catalytic performance of catalysts for HER based on the Sabatier principle.¹¹ Different from the traditional bimetallic and ternary metallic catalysts,^{14,22,26} the active sites of HEA are more diverse and their ΔG_{H^*} values follow a Gaussian distribution, rather than a definite value. Hence, the Sabatier principle and the criterion of $\Delta G_{H^*} = 0$ eV are no longer valid for HEA catalysts. In this work, we propose an anomalous Sabatier principle, where the Gaussian distribution of ΔG_{H^*} with a μ value closer to 0 eV and a larger σ value on HEA catalysts could be used as the descriptor of the higher catalytic activity for HER.”

Report of Reviewer 3

Q. *Based on the PtFeCoNiCu HEA system, this work proposed a “anomalous Sabatier principle” to circumvent the limitations entailed by the well-established Sabatier principle. Specifically, authors state that “Gaussian distribution [$X \sim N(\mu, \sigma^2)$] of ΔG_{H^*} on HEA” could be considered as a descriptor, where a larger σ value results with $\mu = 0$ eV results in a higher catalytic activity for HER. It is indeed a novel idea for explaining the excellent electrocatalytic performance of high-entropy materials derived from the sophisticated “cocktail effect”. However, serious shortcomings and doubtful points exist, which make this paper not ready for being published.*

R: Thanks for the reviewer’s positive comments. We have carefully revised the manuscript according to the reviewer’s comments/suggestions one by one.

Q1. *No in situ experimental results which can remarkably support the authors’ hypothesis were provided in this work. The authors state that “surface Pt sites with strong H^* adsorption are active centers for the Volmer reaction while the ones with weak H^* adsorption are active centers for the Tafel or Heyrovsky reaction”, but it is known that the sites with strong H^* adsorption are adverse to the desorption of adsorbates and thus how could the diffusion of hydrogen from sites with strong H^* adsorption to the sites with weak H^* adsorption be favorable for the overall reaction?*

R1: Thanks for the reviewer’s comments. We have added in situ hydrogen desorption kinetic experiment, which supports the H^* spillover phenomenon on HEA. To determine the hydrogen desorption kinetics, CV tests are performed from 0.1 ~ 0.6 V (vs. RHE) to observe the hydrogen desorption peaks (Nat. Commun., 2021, 12, 3502). As shown in Supplementary Figs. 45a-b, the hydrogen desorption peaks shift with increasing scan rates for both Pt/C and HEA-400-5000. Therefore, their hydrogen desorption kinetics can be quantified through comparing the slopes between the hydrogen desorption peak position vs. scan rates. As shown in Supplementary Fig. 45c, HEA-400-5000 presents a significantly reduced slope than Pt/C, demonstrating the improved

hydrogen desorption kinetics. Previous studies reported that the H* spillover can effectively accelerate the hydrogen desorption kinetics for electrocatalysts (Angew. Chem. Int. Ed., 2019, 58, 16038-16042; Nat. Commun., 2021, 12, 3502; Nat. Commun., 2022, 13, 1189). Thus, the accelerated hydrogen desorption kinetics for HEA-400-5000 could derive from the H* spillover phenomenon on the surface.

Our DFT results indicate that surface Pt sites with strong H* adsorption are active centers for the Volmer reaction (see Fig. 2b-c), and then the adsorbed H* diffuses to the other Pt sites (weak H* adsorption) with the barrier smaller than 0.23 eV (see Fig. 2d-e). Finally, the H₂ is generated on the Pt sites with weak H* adsorption, as shown in Fig. 2b-c. The reaction process without H* diffusion is also considered, as shown in Supplementary Fig. 9. The reaction free energy of potential limiting step for Volmer-Heyrovsky mechanism is 0.099 eV and the reaction barrier for Volmer-Tafel mechanism is 0.519 eV on the strong adsorption site, which are larger than those with H* diffusion. It means that H* diffusion is favorable on our designed HEA system.

To further provide evidence of H* spillover phenomenon on HEA during the HER process, comprehensive electrochemical experiments are conducted. According to previous reports (Nat. Commun., 2022, 13, 1189; Nat. Commun., 2020, 11, 4773), the H* spillover phenomenon can be intuitively observed through the color change in the mixture of catalyst and WO₃ during the HER process. As shown in Supplementary Fig. 42, the colors of WO₃ and Pt/C@WO₃ remain unchanged after an 1800-s HER test (the HER processes are presented in Supplementary Fig. 43). However, the color of HEA-400-5000@WO₃ changed from dark yellow to dark blue after only a 170-s HER test. This is because the spilled-over hydrogen migrates on HEA surface and readily reacts with WO₃ to form dark blue H_xWO₃ (Nat. Commun., 2022, 13, 1189; J. Phys. Chem., 1964, 68, 411-412). The above color change confirms the H* spillover phenomenon on HEA, while no H* spillover occurs on Pt/C. Besides, the H* spillover phenomenon on HEA is verified through the pH-dependent relation of HER. Based on previous reports (Nat. Commun., 2016, 7, 12272; Adv. Funct. Mater., 2017, 27, 1700359; Nat. Commun., 2021, 12, 3502), the H* spillover based HER catalyst possesses a theoretical reaction order of 2.0 between log |j| and pH value. In our work, at an overpotential of -0.05 V (vs. RHE, see Supplementary Fig. 44a), the pH-dependent reaction order for HEA-400-5000 is measured to be 2.03 (see Supplementary Fig. 44b), which is in accord with the theoretical value of 2.0, thus confirming the H* spillover phenomenon. In conclusion, both DFT and experimental results prove the rationality of our proposed mechanism. More details

are shown in the revised manuscript.

“For V-H mechanism, the reaction free energy values of PLS are 0.075 eV and 0.099 eV on the weak and strong adsorption sites, respectively. It means that the overpotential values are 0.075/0.099 V (*vs.* RHE) on weak/strong adsorption sites, which are much smaller than that (0.375 V *vs.* RHE) of Pt (111). When H* spillover is considered, all the electrochemical steps become spontaneous reactions, indicating the better catalytic activity for HER. Similarly, for V-T mechanism, the reaction free energy value of the PLS is 0.075 eV on the weak adsorption site, which will become spontaneous electrochemical reaction with considering H* spillover. For the case on the strong adsorption site, the energy barrier is 0.519 eV for the formation of H₂, which is larger than that (0.297 eV) with H* spillover. Moreover, the H* spillover is much easier on our designed HEA systems with the smaller energy barrier of ~ 0.23 eV than those (0.45 ~ 1.19 eV) on other reported systems where H* spillover proved to occur.²⁸⁻³⁰ It means that the H* spillover should further improve the catalytic activity of HEA systems.” in lines 2-14 of page 5.

“The H* spillover phenomenon can be intuitively observed through the color change in the mixture of catalyst and WO₃ during the HER process. As shown in Supplementary Fig. 42, the colors of WO₃ and Pt/C@WO₃ remain unchanged after an 1800-s HER test (the HER processes are presented in Supplementary Fig. 43). However, the color of HEA-400-5000@WO₃ changed from dark yellow to dark blue after only a 170-s HER test. This is because the spilled-over hydrogen migrates on HEA surface and readily reacts with WO₃ to form dark blue H_xWO₃.^{36,37} The above color change confirms the H* spillover phenomenon on HEA, while no H* spillover occurs on Pt/C. Besides, the H* spillover phenomenon on HEA is verified through the pH-dependent relation of HER. Based on previous reports, the H* spillover based HER catalyst possesses a theoretical reaction order of 2.0 between log |*j*| and pH value.^{29,38,39} At an overpotential of -0.05 V (*vs.* RHE, see Supplementary Fig. 44a), the pH-dependent reaction order for HEA-400-5000 is measured to be 2.03 (see Supplementary Fig. 44b), which is in accord with the theoretical value of 2.0, thus confirming the H* spillover phenomenon. In addition, *in situ* hydrogen desorption kinetics is studied to further support the H* spillover phenomenon on HEA. To determine the hydrogen desorption kinetics, CV tests are performed from 0.1 ~ 0.6 V (*vs.* RHE) to observe the hydrogen desorption peaks.²⁹ As shown in Supplementary Fig. 45a-b, the hydrogen desorption peaks shift with increasing scan rates for both Pt/C and HEA-400-5000. Therefore, their hydrogen desorption kinetics can be quantified through comparing the slopes between the hydrogen desorption peak position *vs.* scan rates. As shown in Supplementary Fig. 45c, HEA-400-5000 presents a significantly reduced slope than Pt/C, demonstrating the improved hydrogen desorption kinetics. Previous study reported that the H* spillover effect can effectively accelerate the hydrogen desorption kinetics for electrocatalysts.^{29,36,40} Thus, the accelerated hydrogen desorption kinetics for HEA-400-5000 could derive from the H* spillover phenomenon on the surface.” in lines 11-34 of page 8.

Supplementary Fig. 45. (a) CV profiles of HEA-400-5000 with the scan rate from 65 to 850 mV s⁻¹ in 0.5 M H₂SO₄. (b) CV profiles of Pt/C with the scan rate from 65 to 850 mV s⁻¹ in 0.5 M H₂SO₄. (c) Plots of hydrogen desorption peak position vs. scan rates for HEA-400-5000 and Pt/C.

Supplementary Fig. 42. Color change photographs of (a) WO₃, (b) mixture of Pt/C@WO₃ and (c) mixture of HEA-400-5000@WO₃. The photographs are taken before and after the HER process as shown in Supplementary Fig. 43.

Supplementary Fig. 43. HER galvanostatic plots for WO₃, Pt/C@WO₃ and HEA-400-5000@WO₃ to identify the spillover mechanism.

Fig. 2. Reaction mechanism studies by DFT calculations. (b) Volmer-Heyrovsky mechanism of HER on 5.9%-HEA (111) and Pt (111). (c) Volmer-Tafel mechanism of HER on 5.9%-HEA (111) and Pt (111). (d) The H* spillover on DR1 (diffusion region for the first H*) for 5.9%-HEA (111). (e) The H* spillover on DR2 (diffusion region for the second H*) for 5.9%-HEA (111).

Supplementary Fig. 9. (a) Volmer-Heyrovsky mechanism of HER on 5.9%-HEA (111) without H* spillover. (b) Volmer-Tafel mechanism of HER on 5.9%-HEA (111) without H* spillover.

Supplementary Fig. 44. (a) Tafel curves of HEA-400-5000 in H₂SO₄ with pH ranging from 0 to 0.4. (b) The liner plot of $\log |j|$ at -0.05 V (vs. RHE) vs. pH for HEA-400-5000.

Q2. *The universality of authors' viewpoint is limited. First, the HEA is generally a system of solid solution with various metal atoms randomly distributed, which means that the atoms on the surface are arranged in various configurations. As the authors state that "the adsorption sites considered in DFT calculations are very limited relative to those on the HEA surface", a particular configuration cannot represent the whole system. Second, proposing an untraditional concept with a new descriptor by only one system is obviously not convincing. Whether the HEA system "with a μ value closer to 0 eV and a larger σ value" but without Pt could also reach the similar electrocatalytic activity to PtFeCoNiCu is doubtful.*

R2: Thanks very much for the reviewer's comments. We strongly agree with the reviewer's idea "A particular configuration cannot represent the whole HEA system". Therefore, we randomly selected more than 400 active sites to calculate ΔG_{H^*} on HEA surfaces with different strains. We found that the ΔG_{H^*} on HEA follows the Gaussian distribution. Therefore, the catalytic performance of HEA should be demonstrated by the fitting Gaussian distribution of ΔG_{H^*} , rather than a particular ΔG_{H^*} on the particular adsorption site. In practice, the active centers for the Volmer (Tafel/Heyrovsky) steps should have stronger (weaker) H^{*} adsorption, respectively, which indicates better catalytic activities of HEAs. Undeniably, the proposed new descriptor is based on PtFeCoNiCu systems. Similar to traditional Sabatier principle, the proposed anomalous Sabatier principle is only related to the adsorption properties of the catalysts, and is not directly related to a particular metal. For example, an HEA system "with a μ value closer to 0 eV and a larger σ value" indicates that there are active sites with stronger (weaker) adsorption for Volmer (Heyrovsky/Tafel) step of HER, resulting in a high catalytic performance for HER, even without Pt in the HEA system. To be fair, this descriptor is valid only for HER on HEA catalysts, while other reactions should

have different μ values.

Q3. *The synthesized materials cannot well match computational model and the proposed concept.*

(1) *For Fig. 3C, the so-called “HEA-core” region is indeed far from the left edge, but it is also located at the bottom edge, and why this region was marked as “core”. Furthermore, we have carefully checked the interplanar spacings in this image based on the scale bar, and the “HEA-surface”, “HEA-transition layer”, and “HEA-core” were measured to be 0.28, 0.25, and 0.25 nm, obviously different from the values provided by authors (0.226, 0.217, and 0.210 nm).*

R3 (1): We are sorry for the confusion on the interplanar spacings. The scale bar has been recalibrated in Fig. 3c and the raw HRTEM-STEM image of HEA-400-5000 is provided for reference. The lattice spacings of HEA-400-5000 in Fig. 3c are specifically reflected in the intensity line profile (for the framed area) as shown in Fig. 3d and Supplementary Fig. 24. The marked lattice spacings in Fig. 3c are actually the 2nd, 11th and 22nd lattice spacings in the intensity line profile which are exactly 0.226, 0.217 and 0.210 nm, respectively (see Supplementary Fig. 24).

As for the lattice naming in Fig. 3c, the three marked lattices are named according to their lattice spacings rather than their locations. It is known that the sample for HRTEM-STEM characterization undergoes powerful ultrasound crushing and peeling, and the selected area for taking images is typically the tiny particles, which were crushed and peeled. Therefore, it lacks evidence to determine whether the lattice belongs to surface or core based on its position in a HRTEM-STEM photo. In Fig. 3c, we named the three marked lattice spacings “HEA-surface”, “HEA-transition layer” and “HEA-core” because their lattice spacings are 0.226, 0.217 and 0.210 nm, respectively. The lattice spacing of 0.226 nm corresponds to that of Pt (111), which indicates that Fe, Co, Ni and Cu elements are etched in this region. The etching effect typically occurs on the surface. Thus, the region with lattice spacings of 0.226 nm is named as HEA-surface. The lattice spacing of 0.210 nm corresponds to that of HEA-400 (111) without CV test, which indicates that no etching effect occurs in this region. In the presence of etching effects (the lattice spacing of 0.226 nm has been detected), the region without etching effect is named as HEA-core.

Fig. 3. (c) HAADF-STEM image of HEA-400-5000. (d) Intensity line profile from the surface to the core in HEA-400-5000 as framed in (c).

Supplementary Fig. 24. Intensity line profile from the surface to the core in HEA-400-5000 as framed in Fig. 3c. The marked 3 lattice spacings correspond to the HEA-surface, HEA-transition layer and HEA-core in Fig. 3c.

Fig. Raw HAADF-STEM image of HEA-400-5000 for Fig. 3c.

(2) *No obvious composition gradients were observed in Fig. 3E, and the mixed figure of PtFeCoNiCu do not match the mapping images of individual Pt, Fe, Co, Ni, and Cu.*

R3 (2): We are sorry for the confusion on the unmatched mixed figure of PtFeCoNiCu in Fig. 3e. The mixed HAADF-EDS elemental maps of HEA-400-5000 which matches the individual Pt, Fe, Co, Ni and Cu, has been provided in the revised manuscript. The HAADF-EDS elemental maps of HEA-400-5000 demonstrate the homogeneously distributed elements in the nanoparticle. To confirm the component gradient in HEA-400-5000, two areas in the near-surface of HEA-400-5000 are selected for further research, as circled in the original image (see Supplementary Fig. 25). As shown in Supplementary Figs. 26-27, Fe, Co, and Ni are less detected than Cu and Pt, confirming the component gradient in the near-surface of HEA-400-5000. The details are shown in lines 2-5 of page 7.

“Moreover, the Pt concentration gradient is further confirmed in the near-surface elemental maps (see Supplementary Figs. 25-27), where Fe, Co, and Ni are less detected than Cu and Pt, due to the susceptibility to corrosion of Fe, Co, and Ni during the activation process.”

Fig. 3e. HAADF-EDS elemental maps of HEA-400-5000.

Supplementary Fig. 25. Origin HAADF TEM image of Fig. 3e. The circled areas I and II are selected for detailed analysis, as shown in Supplementary Figs. 26-27, respectively.

Supplementary Fig. 26. HAADF-EDS elemental maps of HEA-400-5000 selected in the near-surface area I in Supplementary Fig. 25.

Supplementary Fig. 27. HAADF-EDS elemental maps of HEA-400-5000 selected in the near-surface area II in Supplementary Fig. 25.

(3) *The degree of compressive strain in HEA-300, HEA-400, and HEA-500 were not quantitatively analyzed.*

R3 (3): As suggested, the quantitative analysis on the degree of compressive strain has been provided in the revised manuscript. As shown in Supplementary Fig. 12, HEA-300, HEA-400, and HEA-500 have lattice spacings of 0.213, 0.210, and 0.207 nm, corresponding to their (111) planes, respectively. Based on the standard lattice spacing of 0.226 nm for Pt (111) plane (JCPDS 70-2431), HEA-300, HEA-400 and HEA-500 show compressive strains of 5.8%, 7.1% and 8.4%, respectively, as described in lines 37-39 of page 6.

“Compared with the Pt (111) plane (0.226 nm, see Supplementary Fig. 13), HEA-300, HEA-400 and HEA-500 show compressive strains of 5.8%, 7.1% and 8.4%, respectively.”

Supplementary Fig. 12. HRTEM images of (a) HEA-300, (b) HEA-400, and (c) HEA-500.

(4) As shown in Fig. 1, authors show that the μ of ΔG_{H^*} could be adjusted by strain, but why the comparison among HEA-300, HEA-400, and HEA-500 was not displayed, especially the HEA-300?

R3 (4): According to reviewer's comments, the comparison among HEA-300, HEA-400, and HEA-500 has been displayed in the last five lines of page 5 and the first line of page 6 and Supplementary Fig. 14.

“Based on the DFT results, the PtFeCoNiCu with a compressive strain of 6.8% should have a μ value closer to 0 eV, indicating a higher catalytic activity. In agreement with the DFT result, the experimental results indicate HEA-400 with 7.1% strain has a better catalytic performance than HEA-300 (5.8% strain) and HEA-500 (8.4% strain), as shown in Supplementary Fig. 14. Therefore, the system of HEA-400 is our main research object in the following study.”

Supplementary Fig. 14. Polarization curves of HEA catalysts annealed at different temperatures.

(5-1) The electrocatalytic activity of PtFeCoNiCu was significantly boosted and individual Pt nanoparticles appeared after the non-noble metals were etched during CV (Fig. S18).

R3 (5-1): Thanks for the reviewer's comments. To investigate the surface state of HEA-400 after different cycles of CV test, HRTEM images of HEA-400-2000, HEA-400-5000 and HEA-400-10000 have been analyzed and discussed. After 2000 cycles of CV test, HEA-400-2000 presents a (111) lattice spacing of 0.210 nm (see Supplementary Fig. 19), corresponding to that of HEA-400 without CV test. This means that 2000 cycles of CV test are insufficient to introduce component gradient in HEA-400. After 5000 cycles of CV test, besides the original lattice spacing of HEA-400 (0.209 nm, see Area I in Supplementary Fig. 20a), the lattice spacings of activated surface can also be observed, as shown in Area II and Area III in Supplementary Fig. 20a. As shown in Supplementary Fig. 20b, in the activated HEA-surface, the average (111) lattice spacing is 0.227 nm, corresponding to that of Pt (111) lattice. It means that Pt element dominates in the activated surface layer of HEA-400-5000. In the transition layer, the average (111) lattice spacing is 0.216 nm, which is attributed to the partial etching of non-Pt elements. In the core of activated HEA-400-5000, the average (111) lattice spacing is 0.210 nm, which corresponds to that of HEA-400 without CV test, indicating that the etching effect only appears in the outermost layers of HEA. The gradually decreased average lattice spacing from the surface to the core demonstrates that the component gradient rather than the Pt particles exists in HEA-400-5000. Similarly in Area III, the average lattice spacings of HEA-surface and HEA-core are 0.227 and 0.217 nm, respectively, which further supports the existence of component gradient in HEA-400-5000. After 10000 cycles of CV test, HEA-400-10000 presents the lattice spacings of 0.211 nm for HEA (111) and 0.226 nm for Pt (111), respectively (see Supplementary Fig. 21). This result indicates that extra CV tests will lead to excessive etching of non-noble elements in HEA-surface, which will reduce the component gradient. As a result, Pt nanoparticles appear in HEA-400-10000. The details are shown in the revised manuscript and supplementary material.

“To investigate the surface state of HEA-400 after different cycles of CV test, HRTEM characterizations are performed and analyzed. 2000 cycles of CV test are insufficient to introduce component gradient in HEA-400, as shown in Supplementary Fig. 19. After 5000 cycles of CV test, a component gradient appears in HEA-400-5000 (see Supplementary Fig. 20 and Supplementary Discussion). After 10000 cycles of CV test, small Pt particles appear in HEA-400-10000 (see Supplementary Fig. 21). This result indicates that extra CV tests will lead to excessive etching of non-noble elements in HEA-surface, which will reduce the component gradient.” in lines 18-25 of page 6.

“After 2000 cycles of CV test, HEA-400-2000 presents a (111) lattice spacing of 0.210 nm,

corresponding to that of HEA-400 without CV test (see Supplementary Fig. 12b). This means that 2000 cycles of CV test are insufficient to introduce component gradient in HEA-400.” Supplementary Note 1.

“After 5000 cycles of CV test, besides the original lattice spacing of HEA-400 (0.209 nm, see Area I in Supplementary Fig. 20a), the lattice spacings of activated surface can also be observed, as shown in Area II and Area III in Supplementary Fig. 20a. As shown in Supplementary Fig. 20b, in the activated HEA-surface, the average (111) lattice spacing is 0.227 nm, corresponding to that of Pt (111) lattice. It means that Pt element dominates in the activated surface layer of HEA-400-5000. In the transition layer, the average (111) lattice spacing is 0.216 nm, which is attributed to the partial etching of non-Pt elements. In the core of activated HEA-400-5000, the average (111) lattice spacing is 0.210 nm, which corresponds to that of HEA-400 without CV test, indicating that the etching effect only appears in the outermost layers of HEA. This is also the reason for the observed lattice spacing of 0.209 nm which belongs to the original HEA-400 (as framed in Area I). The gradually decreasing average lattice spacing from the surface to the core demonstrates that the component gradient exists in HEA-400-5000. Similarly in Area III, the average lattice spacings of HEA-surface and HEA-core are 0.227 and 0.217 nm, respectively, which further supports the existence of component gradient in HEA-400-5000.” Supplementary Note 2.

“After 10000 cycles of CV test, HEA-400-10000 presents the lattice spacings of 0.211 nm for HEA (111) and 0.226 nm for Pt (111), respectively. This result indicates that extra CV test will lead to excessive etching of non-noble elements in HEA-surface, which will reduce the component gradient.” Supplementary Note 3.

Supplementary Fig. 19. (a) HRTEM image of HEA-400-2000. (b) Intensity line profile of HEA-400-2000 as framed in Area I.

Supplementary Fig. 20. (a) HRTEM image of HEA-400-5000, the inset shows the intensity line profile as framed in Area I. (b) Intensity line profile of HEA-400-5000 as framed in Area II. (c) Intensity line profile of HEA-400-5000 as framed in Area III.

Supplementary Fig. 21. (a) HRTEM image of HEA-400-10000. (b) Intensity line profile of HEA-400-10000 as framed in Area I. (c) Intensity line profile of HEA-400-10000 as framed in Area II.

(5-2) *Whether the boosted activity is attributed to the newly formed Pt NPs, and whether the PtFeCoNiCu is really outperforms the state-of-the-art catalyst is not clear.*

R3 (5-2): The boosted HER activity of HEA-400-5000 derives from the significantly increased ECSA which benefits from the component gradient formed after the CV activation, as confirmed in Supplementary Fig. 33. While the HER activity of HEA-400-10000 is inferior to HEA-400-5000, which is attributed to the diminished component gradient in the surface after excessive CV tests. This result indicates the component gradient rather than the Pt particles playing a role in

enhancing the HER activity. When comparing with Pt/C, the HEA-400-5000 presents smaller overpotential and 4.6 times higher specific electrochemical activity (see Fig. 4c-e), confirming the outperforming activity.

Supplementary Fig. 33. (a) Polarization curves of HEA-400 with different cycles of CV activation. (b) Plots of capacitive currents with various scan rates for HEA-400 with different cycles of CV activation. (c) Polarization curves normalized to ECSA for HEA-400 with different cycles of CV activation.

Fig. 4. (c) Polarization curves of activated HEA-400, activated HEA-500, and Pt/C. The corresponding Tafel plots and exchange current density (j_0) are described in the inset. (d) Specific activities of activated HEA-400, activated HEA-500, and Pt/C at $\eta = 20$ mV. (e) TOFs of activated HEA-400, activated HEA-500, and Pt/C at $\eta = 20$ mV.

(5-3) *If a HEA system without noble metals also displays the similar electrochemical behavior?*

R3 (5-3): To judge whether the electrochemical activation behavior exists in HEA systems without noble metals, a FeCoNiCuMn HEA and a FeCoNiCrMn HEA were synthesized and electrochemically tested using the same CV Potential range (100 ~ 530 mV vs. RHE). As shown in Supplementary Fig. 38, no activation behavior is present in FeCoNiCuMn or FeCoNiCrCu, which indicates that the electrochemical activation behavior may exist only in a HEA system with specific active sites, which possess excellent acid corrosion resistance. The details are shown in the revised manuscript in lines 39-42 of page 7.

“In addition, the HER performance of HEA systems without noble metals are measured. As shown in Supplementary Fig. 38, no activation behavior is present in FeCoNiCuMn or FeCoNiCrCu, which indicates that the electrochemical activation behavior may exist only in a HEA system with specific active sites, which possess excellent acid corrosion resistance.”

Supplementary Fig. 38. (a) Polarization curves of a FeCoNiCuMn HEA with different cycles of CV test. (b) Polarization curves of a FeCoNiCrMn HEA with different cycles of CV test.

(5-4) *The XRD pattern of the PtFeCoNiCu after CV should be provided.*

R3 (5-4): Following reviewer’s suggestion, the XRD patterns of HEA-400-initial, HEA-400-100, HEA-400-2000, HEA-400-5000 and HEA-400-10000 are provided in Supplementary Fig. 23a. After CV activation, all impurity peaks disappeared, indicating that the activated HEA-400 samples are well-defined HEAs. More details are shown in the revised manuscript and supplementary material.

“To further analyze the phase of HEA-400 after different cycles of CV test, XRD characterizations are performed, as shown in Supplementary Fig. 23a. HEA-400-initial presents some impurity peaks corresponding to Co or CoFe species. After 100 cycles of CV test, all the peaks of Co or CoFe species disappear, indicating that HEA-400-100 is a well-defined HEA. The Co and CoFe species that fail to form HEA are etched first during the CV process. In addition to HEA-400-100, HEA-400-2000, HEA-400-5000 and HEA-400-10000 all present well-defined HEA peaks.” in lines 28-34 of page 6.

“HEA-400-initial presents three main peaks at 42.3°, 49.3° and 72.1°, corresponding to the (111), (200) and (220) planes of HEA. The peaks detected at 44.2° and 51.5° correspond to the (111) and (200) planes of Co (JCPDS 89-7093), the peak detected at 44.8° corresponds to the (110) plane of CoFe species (maybe CoFe, JCPDS 44-1433 or Co₃Fe₇ JCPDS 48-1816). After 100 cycles of CV test, all the peaks of Co or CoFe species disappear, indicating that HEA-400-100 is a well-defined HEA. The Co and CoFe species that fail to form HEA are etched first during the CV process. In addition to HEA-400-100, HEA-400-2000, HEA-400-5000 and HEA-400-10000 all present well-defined HEA peaks. Although small Pt nanoparticles have been formed in HEA-400-10000 (see

Supplementary Fig. 21), no Pt characterization peaks are detected in HEA-400-10000, which is attributed to the small amount of the formed Pt particles compared to the whole HEA. The broad peaks detected at $\sim 23.4^\circ$ in HEA catalysts correspond to the glass slide. The appearance of these peaks is because that the amounts of samples collected after HER testing is small, which is not sufficient to fully cover the glass slide, as shown in Supplementary Fig. 23b.” Supplementary Note 4.

Supplementary Fig. 23a. XRD patterns of HEA-400-initial, HEA-400-100, HEA-400-2000, HEA-400-5000 and HEA-400-10000.

(6) *The HEA with individual Pt NPs not match the computational model in Fig. 1A.*

R3 (6): For the main research object of HEA-400-5000 in this work, component gradient rather than the Pt particles exists in the HEA, which matches the computational model. For more details, please see **R3 (5-1)**.

(7) *In Fig. 3A, obvious peak at around 43° appear which is very close to the (111) plane of Cu. Further, in Fig. S24B, the atomic fraction of Cu even exceeds 50%. Thus, the synthetic method in this paper could not obtain the well-defined HEA materials.*

R3 (7): Fig. 3a presents the XRD patterns of HEA samples without CV activation. Admittedly, there are some impurity peaks around 43° . However, after CV activation, all impurity peaks disappeared, indicating that the activated HEA-400 samples are well-defined HEAs (please refer to the corresponding discussions in R3 (5-4) and Supplementary Fig. 23a).

The excessive Cu atomic fraction in Supplementary Fig. 28b (original Supplementary Fig. 24b) derives from the equipment factors during the HAADF-STEM characterization. To conduct the HAADF-STEM characterization, the sample needs to be pretreated through dispersing on a Mo

support film, which is held by a copper specimen holder. The holder is then sent into a vacuum environment for the characterization. In the vacuum environment, the EDS detection signals are easily affected by the surrounding environment. Due to the presence of Cu element in the specimen holder, the detected Cu element content in the sample will inevitably be higher than the actual situation. To support this issue, HAADF line scan characterizations of the blank area without samples are conducted. As shown in Supplementary Fig. 29, a large number of Cu elements are detected due to the equipment factors, even if there are no samples. Therefore, only Fe, Co, Ni and Pt elements are discussed in Supplementary Fig. 28c. In addition, accurate atomic ratios in HEA-400 and HEA-400-5000 are also tested through inductively coupled plasma characterization. For HEA-400 and HEA-400-5000, Pt, Fe, Co, Ni, and Cu elements follow atomic ratios of 18.8: 19.9: 22.3: 20.2: 18.8 and 35.0: 12.3: 14.8: 17.3: 20.6, respectively. More details are shown in the revised manuscript and supplementary material.

“Taking HEA-400 as an example, Pt, Fe, Co, Ni, and Cu elements follow an atomic ratio of 18.8: 19.9: 22.3: 20.2: 18.8, according to inductively coupled plasma optical emission spectroscopy (ICP-OES)” in lines 13-15 of page 6.

“The Pt, Fe, Co, Ni and Cu elements in HEA-400-5000 follow an atomic ratio of 35.0: 12.3: 14.8: 17.3: 20.6 according to ICP-OES results, consistent with our assumption of the dissolution of Fe, Co, Ni, and Cu components on the surface during the CV activation.” in lines 17-20 of page 7.

“The excessive Cu atomic fraction in Supplementary Fig. 28b derives from the equipment factors during the HAADF-STEM characterization. To conduct the HAADF-STEM characterization, the sample needs to be pretreated through dispersing on a Mo support film, which is held by a copper specimen holder. The holder is then sent into a vacuum environment for the characterization. In the vacuum environment, the EDS detection signals are easily affected by the surrounding environment. Due to the presence of Cu element in the specimen holder, the detected Cu element content in the sample will inevitably be higher than the actual situation. To support this issue, HAADF line scan characterizations of the blank area without samples are conducted. As shown in Supplementary Fig. 29, a large number of Cu elements are detected due to the equipment factors, even if there are no samples.” Supplementary Note 5.

Supplementary Fig. 28. (a) HAADF image of HEA-400-5000. (b) HAADF line scan of Fe, Co, Ni, Cu, and Pt elements along the green arrow shown in Supplementary Fig. 28a. (c) HAADF line scan of Fe, Co, Ni, and Pt elements along the green line shown in Supplementary Fig. 28a. Note that inevitable errors exist in the data of the Cu element due to the equipment factors. Thus, they are removed for clarity.

Supplementary Fig. 29. (a) HAADF image of HEA-400-5000. The white areas are the HEA particles, the black area is blank. (b) HAADF line scan of Fe, Co, Ni, Cu, and Pt elements along the green arrow shown in Area 1. (c) HAADF line scan of Fe, Co, Ni, Cu, and Pt elements along the green arrow shown in Area 2.

(8) Whether the individual Cu phase disappear after CV is very suspicious. In Fig. S22-23, the distribution of Cu and Pt is quite similar and authors state that “Fe, Co, and Ni are less detected than Cu and Pt, due to the susceptibility to corrosion of Fe, Co, and Ni during the activation process”. However, in Fig. 24C, the line scan curve of Cu was hidden due to the “inevitable errors” caused by the “equipment factors”, which is quite suspicious.

R3 (8): Please see R3 (7).

Report of Reviewer 4

In this paper, Chen, Li et al. proposed the anomalous Sabatier principle on high entropy alloys electrocatalysts for hydrogen evolution reaction, where catalytic activity is based on the Gaussian distribution of ΔG_{H^} (influenced by composition, strain, and synthesis conditions). Additionally, those theoretical findings were confirmed by precisely designed catalyst synthesis and experiments. In my opinion, these findings are of high importance for developing high entropy catalysts. Therefore, I recommend the publication of this work in Nature Communication after addressing in the revised version the following points:*

R: Thanks for the reviewer's positive comments. We have revised the manuscript according to the reviewer's comments/suggestions one by one.

Q1. *Fig.2. what exactly are DR1 and DR2? Diffusion regions for active sides with strong and weak H bonding?*

R1: We are sorry for the confusion. DR1 and DR2 are the diffusion regions for the first H* and the second H*, respectively. We have added the corresponding description in the figure caption.

Q2. *Temperature used by catalyst synthesis is relatively low. Can authors elaborate more about the mixability and stability of this material system (maybe on the examples of phase diagrams)?*

R2: Following reviewer's suggestion, higher annealing temperatures of 700 and 900 °C are used to synthesize PtFeCoNiCu HEA. The corresponding XRD characterizations are performed. As shown in Supplementary Fig. 56, all the samples present 3 main peaks corresponding to the (111), (200) and (220) planes of a face-centered cubic HEA structure. For HEA-400, three impure phase peaks are detected. The peaks at 44.2° and 51.5° correspond to the (111) and (200) planes of Co (JCPDS 89-7093), the peak at 44.8° corresponds to the (110) plane of CoFe species (maybe CoFe, JCPDS 44-1433 or Co₃Fe₇ JCPDS 48-1816). When increasing the annealing temperature, CoFe phases disappear while Co peaks still exist in HEA-500 and HEA-700. A well-defined HEA is synthesized when further increasing the annealing temperature to 900 °C. The detected (111) planes for HEA-400, HEA-500, HEA-700 and HEA-900 locate at 42.1°, 42.1°, 42.4° and 42.5°, respectively. Positive shifts of the (111) peaks as the annealing temperature increases imply the greater compressive strain in the HEAs. According to DFT results, the HEA with a strain of 6.8% should show a higher catalytic performance for HER. Besides, the HEA particle size also increases

with the annealing temperature, a large particle size is not conducive to an excellent HER activity. Therefore, although the HEA-900 presents a well-defined HEA-structure, the excessive particle size (70 ~ 250 nm, see Supplementary Fig. 57a) and compressive strain (9.7%, based on the standard lattice spacing of 0.226 nm for Pt (111), see Supplementary Fig. 57b) make HEA-900 not the best choice for HER. HEA-400 possesses appropriate particle size (30 ~ 40 nm) and compressive strain (7.1%), and thus serves as the main research object in this work. Besides, after CV activation, all the impure phase peaks disappeared (see Supplementary Fig. 23a), indicating that the activated HEA-400 samples are well-defined HEAs. In addition, the HEA-400-5000 presents almost unchanged potential and morphology after an 80-h galvanostatic measurement (see Supplementary Fig. 31), verifying the high stability. More details are shown in the revised manuscript and supplementary material.

“Besides, the high stability of HEA-400-5000 is verified through the galvanostatic measurement during an 80-h test (see Supplementary Fig. 31).” in lines 25-27 of page 7.

“All the samples present 3 main peaks corresponding to the (111), (200) and (220) planes of a face-centered cubic HEA structure. For HEA-400, three impure phase peaks are detected. The peaks at 44.2° and 51.5° correspond to the (111) and (200) planes of Co (JCPDS 89-7093), the peak at 44.8° corresponds to the (110) plane of CoFe species (maybe CoFe, JCPDS 44-1433 or Co₃Fe₇ JCPDS 48-1816). When increasing the annealing temperature, CoFe phases disappear while Co peaks still exist in HEA-500 and HEA-700. A well-defined HEA is synthesized when further increasing the annealing temperature to 900 °C. The detected (111) planes for HEA-400, HEA-500, HEA-700 and HEA-900 locate at 42.1°, 42.1°, 42.4° and 42.5°, respectively. Positive shifts of the (111) peaks as the annealing temperature increases imply the greater compressive strain in the HEAs. According to DFT results, the HEA with a strain of 6.8% should show a higher catalytic performance for HER. Besides, the HEA particle size also increases with the annealing temperature, a large particle size is not conducive to an excellent HER activity. Therefore, although the HEA-900 presents a well-defined HEA-structure, the excessive particle size (70 ~ 250 nm, see Supplementary Fig. 57a) and compressive strain (9.7%, based on the standard lattice spacing of 0.226 nm for Pt (111), see Supplementary Fig. 57b) make HEA-900 not the best choice for HER.” Supplementary Note 10.

“HEA-400-initial presents three main peaks at 42.3°, 49.3° and 72.1°, corresponding to the (111), (200) and (220) planes of HEA. The peaks detected at 44.2° and 51.5° correspond to the (111) and (200) planes of Co (JCPDS 89-7093), the peak detected at 44.8° corresponds to the (110) plane of CoFe species (maybe CoFe, JCPDS 44-1433 or Co₃Fe₇ JCPDS 48-1816). After 100 cycles of CV test, all the peaks of Co or CoFe species disappear, indicating that HEA-400-100 is a well-defined HEA. The Co and CoFe species that fail to form HEA are etched first during the CV process. In addition to HEA-400-100, HEA-400-2000, HEA-400-5000 and HEA-400-10000 all present well-defined HEA peaks. Although small Pt nanoparticles have been formed in HEA-400-10000 (see

Supplementary Fig. 21), no Pt characterization peaks are detected in HEA-400-10000, which is attributed to the small amount of the formed Pt particles compared to the whole HEA. The broad peaks detected at $\sim 23.4^\circ$ in HEA catalysts correspond to the glass slide. The appearance of these peaks is because that the amounts of samples collected after HER testing is small, which is not sufficient to fully cover the glass slide, as shown in Supplementary Fig. 23b.” Supplementary Note 4.

Supplementary Fig. 56. XRD patterns of HEA-400, HEA-500, HEA-700 and HEA-900.

Supplementary Fig. 57. (a) TEM image of HEA-900. (b) HRTEM image of HEA-900.

Supplementary Fig. 23a. XRD patterns of HEA-400-initial, HEA-400-100, HEA-400-2000, HEA-400-5000 and HEA-400-10000.

Supplementary Fig. 31. The galvanostatic plot of activated HEA-400 (HEA-400-5000) at -10 mA cm^{-2} for 80 h. The inset shows a TEM image of HEA-400-5000 after the 80-h test.

Q3. As the authors also mentioned, catalysts prepared at various temperatures have significantly different electrochemically active surface areas (ECSA). For clarity of presentation and to prove that the observed increase of activity is not only an effect of increased surface area, linear sweep voltammograms in figures: 4A, 4C, S10, and S25 should also be normalized to the ECSA.

R3: Following reviewer's suggestion, polarization curves normalized to ECSA for HEA-400 after different cycles of CV tests (see Supplementary Fig. 33c, corresponding to Fig. 4a), polarization curves normalized to ECSA for HEA-500 after different cycles of CV tests (see Supplementary Fig. 39c, corresponding to Supplementary Fig. 24), polarization curves normalized to ECSA of HEA-400-5000, HEA-500-2000, and Pt/C (see Supplementary Fig. 41, corresponding to Fig. 4c)

and polarization curves normalized to ECSA for the comparison among HEA-300, HEA-400, HEA-500 and Pt/C (see Supplementary Fig. 58, corresponding to original Supplementary Fig. 10) are provided. The corresponding discussions have been added to the revised supplementary material.

“For HEA-400, the C_{dl} increases continuously with the CV progress and reaches the maximum value of 111.7 mF cm^{-2} after 5000 CV cycles (see Supplementary Fig. 33b). The greatly increased C_{dl} derives from the gradually increasing exposed Pt sites and the enlarged component gradient during the CV test. Further CV tests will reduce the C_{dl} of HEA-400 (the C_{dl} of HEA-400-10000 is 95.0 mF cm^{-2}), which may be attributed to the diminished component gradient in the surface after excessive CV tests. Corresponding to the C_{dl} , the HER activity of HEA-400 continues to improve with the CV progress and HEA-400-5000 shows the best performance with an η_{100} of 30.7 mV (see Supplementary Fig. 33a). Further CV tests make the HER activity of HEA-400 reduced, the HEA-400-10000 shows an increased η_{100} of 36.6 mV . When normalized to the ECSA, the specific activity of HEA-400 declines slightly with the CV progress. This is because the electron gradient in HEA surface gradually decreases during the CV progress.” Supplementary Note 6.

“For HEA-500, the C_{dl} and HER activity increase continuously with the CV progress and reach the best performance after 2000 CV cycles. The HEA-500-2000 show a C_{dl} of 40.5 mF cm^{-2} and an η_{100} of 42.1 mV . Further CV tests will reduce the C_{dl} and HER activity of HEA-500. HEA-500-5000 shows a diminished C_{dl} of 34.9 mF cm^{-2} and an increased η_{100} of 62.5 mV . When normalized to the ECSA, the specific activity of HEA-500 declines slightly with the CV progress, which is also attributed to the diminished electron gradient in the surface.” Supplementary Note 7.

Supplementary Fig. 33c. Polarization curves normalized to ECSA for HEA-400 with different cycles of CV activation.

Supplementary Fig. 39c. Polarization curves normalized to ECSA for HEA-500 with different cycles of CV activation.

Supplementary Fig. 41. Polarization curves normalized to ECSA of HEA-400-5000, HEA-500-2000, and Pt/C.

Supplementary Fig. 58. Polarization curves normalized to ECSA for HEA-300, HEA-400, HEA-500 and Pt/C.

Q4. *Potential range of cyclic voltammetry conducted as an activation process should also be mentioned in the main part of the manuscript.*

R4: Thanks for the reviewer's comments. The potential range of CV conducted for activation is 100 ~ 530 mV (vs. HRE). The corresponding discussion has been added in lines 15-16 of page 6.

“To exclude the impact of impure phase and to bring component gradient in HEA, CV measurements are conducted for activation at a range of 100 ~ 530 mV (vs. RHE).”

Q5. *How can authors be sure that observed currents are only related to hydrogen evolution reaction (HER)? For example, there might be an overlap of the catalytic current of HER and the current resulting from the dissolution/corrosion of non-noble metal elements. Measuring faradaic efficiencies and/or determining the amount of produced hydrogen might be helpful to answer this question.*

R5: Following reviewer's suggestion, the amount of produced H₂ and the corresponding faradic efficiency of HEA-400-5000 were measured. The hydrogen production was measured with a gas chromatograph (GC-2014), using a thermal conductivity detector (TCD) to detect H₂ content every 10 min. For detecting the amount of produced H₂, 1 mg HEA-400-5000 catalyst is loaded on the Ni foam with sufficient Nafion as the binder. During the 50-min test, a total of 0.067 mmol H₂ is produced through 1 mg HEA-400-5000 catalyst under a potential of -0.5 V (vs. RHE, see Supplementary Fig. 32). The corresponding faradic efficiency is calculated to be 99%, demonstrating that almost all of the current in the test is used for HER. The details are shown in the revised manuscript.

“The faradic efficiency of HEA-400-500 is also calculated to be 99% (see Supplementary Fig. 32), demonstrating that almost all of the current in the test is used for HER.” in lines 27-29 of page 7.

“The hydrogen production was measured with a gas chromatograph (GC-2014), using a thermal conductivity detector (TCD) to detect H₂ content every 10 min. For detecting the amount of produced H₂, 1 mg catalyst was loaded on the Ni foam with sufficient Nafion as the binder.” in the last three lines of page 10 and the first line of page 11.

Supplementary Fig. 32. Hydrogen production and the corresponding faradic efficiency for HEA-400-5000.

Q6. *Caption of figure S21: S20 and S21 should be changed to S22 and S23, respectively.*

R6: We are sorry for the unmatched caption provided in Supplementary Fig. 25 (original Supplementary Fig. 21). The corresponding caption has been revised in the revised supplementary material.

Supplementary Fig. 25. Origin HAADF TEM image of Fig. 3e. The circled areas I and II are selected for detailed analysis, as shown in Supplementary Figs. 26-27, respectively.

Q7. *Why was a potential window of 10 mV chosen to determine electrochemically active surface area? It is known from the literature that broader potential windows (100 mV or more) are recommended (see, for example, DOI 10.1088/2515-7655/abee33).*

R7: Thanks for the reviewer's comments. The potential window to determine electrochemically surface area is actually 100 mV, from 100 to 200 mV (vs. RHE), as shown in Supplementary Fig.

34 for HEA-400, Supplementary Fig. 40 for HEA-500 and Supplementary Fig. 47b for Pt/C.

Supplementary Fig. 34. (a-e) Detailed cyclic voltammetry data for HEA-400-initial, HEA-400-100, HEA-400-2000, and HEA-400-5000, HEA-400-10000 to determine the double layer capacitance, respectively.

Supplementary Fig. 40. (a-d) Detailed cyclic voltammetry data for HEA-500-initial, HEA-500-100, HEA-500-2000, and HEA-400-5000 to determine the double layer capacitance, respectively.

Supplementary Fig. 47b. Detailed CV data for Pt/C to determine the double layer capacitance.

Q8. Why in comparison plot in Fig. S31 activation of sample HEA-400 and HEA-500 were different?

R8: For HEA-400, the double-layer capacitance (C_{dl}) increases continuously with the CV progress and reaches the maximum value of 111.7 mF cm^{-2} after 5000 CV cycles (see Supplementary Fig. 33b). Further CV tests will reduce the C_{dl} of HEA-400 (the C_{dl} of HEA-400-10000 is 95.0 mF cm^{-2}). Correspondingly, HEA-400-5000 shows the best performance with an η_{100} of 30.7 mV (see Supplementary Fig. 33a). Therefore, for HEA-400, 5000 CV cycles can bring the best HER performance. While for HEA-500, the C_{dl} and HER activity increase continuously with the CV progress and reach the best performance after 2000 CV cycles (see Supplementary Fig. 39). Further CV tests will reduce the C_{dl} and HER activity of HEA-500. Therefore, for HEA-500, 2000 CV cycles can bring the best HER performance. The trends of HER performance change during the CV test for HEA-400 and HEA-500 are the same. Both of them experience a significant increase followed by a slight decrease in HER performance with continuous CV test. However, the cycles of CV tests to bring the best HER performance for HEA-400 and HEA-500 are different. During the CV activation process, the HEA particles undergo the removal of impurities (as confirmed in the XRD patterns shown in Supplementary Fig. 23a) and the formation of component gradient. More impure phases exist in HEA-400 than HEA-500, as mentioned in **R2**. Therefore, more CV tests are performed to etch these impurities in HEA-400 before bringing the component gradient on the surface. Consequently, more activation cycles are performed to bring the best HER performance for HEA-400 than HEA-500. The details are shown on Supplementary Note 8.

“The trends of HER performance change during the CV test for HEA-400 and HEA-500 are the same. Both of them experience a significant increase followed by a slight decrease in HER performance with continuous CV test. However, the cycles of CV tests to bring the best HER performance for HEA-400 and HEA-500 are different. During the CV activation process, the HEA particles undergo the removal of impurities (as confirmed in the XRD patterns shown in Supplementary Fig. 23a) and the formation of component gradient. More impure phases exist in HEA-400 than HEA-500. Therefore, more CV tests are performed to etch these impurities in HEA-400 before bringing the component gradient on the surface. Consequently, more activation cycles are performed to bring the best HER performance for HEA-400 than HEA-500.”

Supplementary Fig. 33. (a) Polarization curves of HEA-400 with different cycles of CV activation. (b) Plots of capacitive currents with various scan rates for HEA-400 with different cycles of CV activation.

Supplementary Fig. 39. (a) Polarization curves of HEA-500 with different cycles of CV activation. (b) Plots of capacitive currents with various scan rates for HEA-500 with different cycles of CV activation.

REVIEWER COMMENTS

Reviewer #1 (Remarks to the Author):

While I do not doubt the observation that the catalyst found by the authors has enhanced activity compared to Pt/C, I remain unconvinced that the reason for it should be an “anomalous Sabatier principle” caused by surface diffusion of hydrogen (or hydrogen spillover).

I still believe that the results put forward by authors could equally well be explained and predicted by a compressive strain effect of the Pt overlayers in the structure of the proposed catalyst. Compression of the Pt-atoms would lead to weakening of the H* adsorption energies, which presumably would result in a lower overpotential for the HEA catalyst, according to the reaction energy diagram in Figure 2b and c.

A simulation that could shed light on this would be to show that a non-HEA-Pt(111) surface with a reduced lattice constant corresponding to the Pt(111) overlayer on the HEA under compressive strain is actually distinct from the Pt(111) overlayer on the HEA. This computational experiment would tell whether the electron gradients, and hence the proposed “anomalous Sabatier principle”, are relevant or redundant as a descriptor for explaining, and predicting, the increased catalytic activity.

One reason that the compressive strain appears sufficient to explain the experimental observations is that the transition state energy of the Tafel step on the HEA in Supplementary Figure 9b appears to be independent of whether the site on the HEA adsorbs H* weakly or strongly, the transition state energy is unchanged at around 0.35 eV, but yet different from pure Pt(111) at around 0.6 eV (from Figure 2c). This suggests that the transition state energy could be related only to the degree of strain of the Pt(111) overlayer, and hence the strain of the overlayer would be an adequate descriptor of the reactivity.

I acknowledge the ease of the diffusion of adsorbed H* on the surface, and the fact that the adsorbed H* is able to spill over to the WO₃ support as illustrated in Supplementary Figure S42. However, this increased mobility of the adsorbate could also be considered a consequence of the weaker adsorption strength on the HEA compared to Pt/C. It can be speculated that compressively strained non-HEA-Pt(111) overlayers would possess the same spillover properties.

The proposed descriptor of increasing reactivity by having H* adsorption energies centered around 0 is also what an analysis of strained overlayers would suggest, however I remain unconvinced that an increased variance in the adsorption energies around zero adsorption energy should benefit the reactivity. Even if it happened to be the case that a variance in the adsorption energies would increase the reactivity (i.e. if more HEAs were considered on the study), I would suspect a particularly low transition state energy on a particular site or perhaps surface coverage effects to be the reason for it. I would not suspect surface diffusion as a reason (as long as diffusion on the surface is not rate-determining) because of the premise that diffusing from a strongly adsorbing site (a) to a weakly adsorbing site (b), and then to the transition state (c) would happen at the same rate as going from the strongly adsorbing site (a) to the transition state (c) simply because only the

overall energy difference is relevant for the Boltzmann probability of observing the system in the initial and final state.

On a different note, I notice an incorrect author list of reference 40.

Reviewer #2 (Remarks to the Author):

The authors had wonderfully addressed all the concerns raised from the previous submission round. Hence, this work is recommended for publication without any further revision.

Reviewer #3 (Remarks to the Author):

The revision was significantly promoted, and most of the concerns have been well discussed. We consider that the manuscript would be ready for published after addressing the following questions.

1. The H spillover mechanism is reasonable and the activity of PtFeCoNiCu could be well explained by the proposed "Gaussian distribution [$X \sim N(\mu, \sigma^2)$] of ΔGH^* on HEA", and It is indeed a novel idea for explaining the excellent electrocatalytic performance of high-entropy materials derived from the sophisticated "cocktail effect". However, the "HEA system with a μ value closer to 0 eV and a larger σ value but without Pt" was still not supplied. Since the proposed "descriptor" is only effective in case of the "HEA system with specific active sites", the name of "anomalous Sabatier principle" is not accurate.

2. In the revised paper, the sophisticated configuration of PtFeCoNiCu HEA system has been well investigated by randomly selecting more than 400 active sites. For clarity, the resulting ΔGH^* values are recommended to be summarized and partially displayed in SI (for example, Fig. S55 and Table 9 in the article with DOI: 10.1088/2515-7655/abee33)

Reviewer #4 (Remarks to the Author):

The authors answered all my questions. I recommend publishing the revised version of the manuscript.

Detailed Responses to the Comments

Report of Reviewer 1

Q1 (question 1). *While I do not doubt the observation that the catalyst found by the authors has enhanced activity compared to Pt/C, I remain unconvinced that the reason for it should be an “anomalous Sabatier principle” caused by surface diffusion of hydrogen (or hydrogen spillover). I still believe that the results put forward by authors could equally well be explained and predicted by a compressive strain effect of the Pt overlayers in the structure of the proposed catalyst. Compression of the Pt-atoms would lead to weakening of the H* adsorption energies, which presumably would result in a lower overpotential for the HEA catalyst, according to the reaction energy diagram in Figure 2b and c.*

A simulation that could shed light on this would be to show that a non-HEA-Pt(111) surface with a reduced lattice constant corresponding to the Pt(111) overlayer on the HEA under compressive strain is actually distinct from the Pt(111) overlayer on the HEA. This computational experiment would tell whether the electron gradients, and hence the proposed “anomalous Sabatier principle”, are relevant or redundant as a descriptor for explaining, and predicting, the increased catalytic activity.

One reason that the compressive strain appears sufficient to explain the experimental observations is that the transition state energy of the Tafel step on the HEA in Supplementary Figure 9b appears to be independent of whether the site on the HEA adsorbs H weakly or strongly, the transition state energy is unchanged at around 0.35 eV, but yet different from pure Pt(111) at around 0.6 eV (from Figure 2c). This suggest that the transition state energy could be related only to the degree of strain of the Pt(111) overlayer, and hence the strain of the overlayer would be an adequate descriptor of the reactivity.*

R1 (reply 1): Thanks for your insightful comments, which have prompted us to delve further into the underlying mechanism for our findings. We entirely agree with your assessment that compression strain plays a significant role in weakening the adsorption energies of H* on the catalyst surface, thereby leading to a lower overpotential for the HEA catalysts.

In light of your suggestion, we have performed simulations to investigate the HER process on a Pt (111) surface (see Supplementary Fig. 12) with a matching compressive strain of 5.9%, equivalent to that of the HEA (111). The corresponding reaction free energy values are shown in

Supplementary Table 1. In Volmer-Heyrovsky mechanism, the Heyrovsky step becomes the potential-limiting step with a ΔG_{Hey} value of 0.270 eV for 5.9%-Pt (111). It is noteworthy that for the 5.9%-HEA (111), all steps demonstrate an exothermic nature, indicating higher catalytic activity. Similarly, within the Volmer-Tafel mechanism, the rate-determining step shows a ΔG_{Tafel} of 1.006 eV for 5.9%-Pt (111), significantly larger than the corresponding 0.297 eV for 5.9%-HEA (111). This further underscores the catalytic superiority of HEA (111) over Pt (111), even under identical compressive strain conditions. We have also conducted similar simulations on the HEA (111) with the lattice constant of unstrained Pt (111), as shown in Supplementary Fig. 13 and Supplementary Table 1. In this scenario, the Volmer-Heyrovsky mechanism reveals that the potential-limiting step is the Heyrovsky step with a reduced ΔG_{Hey} of 0.197 eV for HEA (111), in contrast to the 0.375 eV for Pt (111). Correspondingly, the Volmer-Tafel mechanism highlights a rate-determining step with a ΔG_{Tafel} of 0.578 eV for HEA (111), much lower than 1.128 eV for Pt (111). These findings collectively demonstrate that, beyond the strain effect, the HEA with the electron gradients possesses intrinsic advantages in enhancing the catalytic activity of HER.

Addressing your point regarding the relative energy of transition states in Supplementary Fig. 9b, we acknowledge that the values for both weak and strong adsorption sites are remarkably consistent. However, it's crucial to emphasize that the energy barriers for H₂ formation differ significantly between the two sites—0.519 eV for the strong adsorption site and 0.297 eV for the weak adsorption site, respectively. This insight underscores the multifaceted nature of factors influencing H₂ formation, extending beyond just strain to encompass the adsorption energies of active sites.

We deeply appreciate your engagement with our work and the valuable discourse that has ensued. Your feedback has been instrumental in refining our understanding of the underlying mechanisms driving enhanced catalytic activity.

“Undoubtedly, the influence of strain effect holds a pivotal significance in the enhancement of catalytic performance within HEA catalysts. Furthermore, our investigation extended to encompass the catalytic process of HER on both the Pt (111) with a compressive strain of 5.9% (Supplementary Fig. 12) and the unstrained HEA (111) (Supplementary Fig. 13). Notably, our comparative analysis reveals that, under the same strains, HEA catalysts consistently exhibit superior catalytic performance when juxtaposed with Pt (111). This discernible disparity can potentially be attributed to the intricate interplay of adsorption energies of H* stemming from the variegated electronic gradients present across the surface of HEA catalysts.” in lines 23-30 of page 5.

Supplementary Fig. 12. Reaction process of HER on 5.9%-HEA (111) and 5.9%-Pt (111). (a) Volmer-Heyrovsky mechanism of HER on 5.9%-HEA (111) and 5.9%-Pt (111). (b) Volmer-Tafel mechanism of HER on 5.9%-HEA (111) and 5.9%-Pt (111).

Supplementary Table 1. Reaction free energy values of all steps during HER on 5.9%-HEA (111), HEA (111), 5.9%-Pt (111) and Pt (111).

Reactions	5.9%-HEA (111)	HEA (111)	5.9%-Pt (111)	Pt (111)
ΔG_{Vol-1} (eV)	-0.099 (-0.109 ^a)	-0.358	-0.270	-0.375
ΔG_{Vol-2} (eV)	-0.091 (-0.087 ^a)	-0.297	-0.114	-0.201
ΔG_{Hey} (eV)	-0.075 (-0.039 ^a)	0.197	0.270	0.375
ΔG_{Tafel} (eV)	0.297 (0.370 ^a)	0.578	1.006	1.128

The symbol of ^a indicates the reaction process of HER on another randomly active center of 5.9%-HEA (111) (see Supplementary Fig. 10).

Supplementary Fig. 13. Reaction process of HER on HEA (111) and Pt (111). (a) Volmer-Heyrovsky mechanism of HER on HEA (111) and Pt (111). (b) Volmer-Tafel mechanism of HER on HEA (111) and Pt (111).

Q2. I acknowledge the ease of the diffusion of adsorbed H^* on the surface, and the fact that the

adsorbed H is able to spill over to the WO₃ support as illustrated in Supplementary Figure S42. However, this increased mobility of the adsorbate could also be considered a consequence of the weaker adsorption strength on the HEA compared to Pt/C. It can be speculated that compressively strained non-HEA-Pt(111) overlayers would possess the same spillover properties.*

The proposed descriptor of increasing reactivity by having H adsorption energies centered around 0 is also what an analysis of strained overlayers would suggest, however I remain unconvinced that an increased variance in the adsorption energies around zero adsorption energy should benefit the reactivity. Even if it happened to be the case that a variance in the adsorption energies would increase the reactivity (i.e. if more HEAs were considered on the study), I would suspect a particularly low transition state energy on a particular site or perhaps surface coverage effects to be the reason for it. I would not suspect surface diffusion as a reason (as long as diffusion on the surface is not rate-determining) because of the premise that diffusing from a strongly adsorbing site (a) to a weakly adsorbing site (b), and then to the transition state (c) would happen at the same rate as going from the strongly adsorbing site (a) to the transition state (c) simply because only the overall energy difference is relevant for the Boltzmann probability of observing the system in the initial and final state.*

R2: Thanks for the reviewer's comments. We agree that weaker H* adsorption would facilitate the diffusion of H* on the compressively strained non-HEA-Pt (111) surface. However, on the homogeneous active sites of Pt (111), this diffusion is inefficient, because the uniformity of surface Pt sites leads to consistent energy barriers for H₂ formation, regardless of the diffusion pathway taken by H*. This dynamic change on the HEA (111) surface is characterized by its diverse adsorption sites. In this context, when strong adsorption sites are adopted as the initial state, the energy barrier for H₂ formation is expected to be greater than that originating from weak adsorption sites. The experimental observations of H* spillover in this work reinforce our belief in the potential role of this mechanism in improving the overall catalytic performance of the hydrogen evolution reaction (HER).

We acknowledge that there may still be some controversies regarding the “anomalous Sabatier principle”, which should be further studied in the future. We have also taken your feedback into consideration and have toned down some claims to ensure a balanced presentation. We hope to convey our understanding that this is a novel concept requiring further exploration and validation. Therefore, more efforts are needed to better understand this issue.

Q3. *On a different note, I notice an incorrect author list of reference 40.*

R3: Thanks very much for pointing out this error. We have revised the author list of reference 40, as shown below:

“40. Park, J., Lee, S., Kim, H. E., Cho, A., Kim, S., Ye, Y., Han, J. W., Lee, H., Jang, J. H. & Lee, J. Investigation of the support effect in atomically dispersed Pt on $\text{WO}_{(3-x)}$ for utilization of Pt in the hydrogen evolution reaction. *Angew. Chem. Int. Ed.* **58**, 16038-16042 (2019).”

Report of Reviewer 2

Q. *The authors had wonderfully addressed all the concerns raised from the previous submission round. Hence, this work is recommended for publication without any further revision.*

R: Many thanks to the reviewer for the previous comments/suggestions, which significantly improved the quality of this manuscript.

Report of Reviewer 3

Q. *The revision was significantly promoted, and most of the concerns have been well discussed. We consider that the manuscript would be ready for published after addressing the following questions.*

R: Thanks for the reviewer’s positive comments. We have carefully revised the manuscript according to the reviewer’s comments/suggestions one by one.

Q1. *The H spillover mechanism is reasonable and the activity of PtFeCoNiCu could be well explained by the proposed “Gaussian distribution [$X \sim N(\mu, \sigma^2)$] of ΔG_{H^*} on HEA”, and It is indeed a novel idea for explaining the excellent electrocatalytic performance of high-entropy materials derived from the sophisticated “cocktail effect”. However, the “HEA system with a μ value closer to 0 eV and a larger σ value but without Pt” was still not supplied. Since the proposed “descriptor” is only effective in case of the “HEA system with specific active sites”, the name of “anomalous Sabatier principle” is not accurate.*

R1: Thanks for the reviewer’s comments. In this work, the proposed descriptor is the Gaussian distribution [$X \sim N(\mu, \sigma^2)$] of ΔG_{H^*} on HEA surfaces. We would like to clarify that this descriptor is linked to the composition and surface characteristics of HEAs, rather than being limited to

specific active sites. We apologize for any misunderstanding that may have arisen from our presentation in Fig. 2, where we focused on illustrating the reaction process of HER at a single randomly selected active center on the 5.9%-HEA (111) surface. To further verify the universality of our proposed descriptor, we have randomly selected an additional active center on the 5.9%-HEA (111) surface and calculated the reaction process of HER, as demonstrated in Supplementary Fig. 10. This analysis reaffirms the high catalytic performance of HER across different active centers on the same HEA surface (Supplementary Table 1). This additional analysis highlights the robustness of our proposed "anomalous Sabatier principle" and its applicability beyond specific active sites. The traditional Sabatier principle requires that the adsorbate should bind neither too weakly nor too strongly. On HEA surface, however, favorable adsorption energies are distinct—strong adsorption sites are required to be stronger, while weak adsorption sites should be weaker. This phenomenon serves to enhance the efficiency of HER. Given your perspective and to ensure precision in our terminology, we have softened the “anomalous Sabatier principle” to “unusual Sabatier principle” and toned down some claims.

“we discovered an unusual Sabatier principle on high entropy alloy (HEA) surface, distinguishing the “just right” ($\Delta G_{H^*} = 0$ eV) in the Sabatier principle of hydrogen evolution reaction (HER)” in the abstract.

Supplementary Table 1. Reaction free energy values of all steps during HER on 5.9%-HEA (111), HEA (111), 5.9%-Pt (111) and Pt (111).

Reactions	5.9%-HEA (111)	HEA (111)	5.9%-Pt (111)	Pt (111)
ΔG_{Vol-1} (eV)	-0.099 (-0.109 ^a)	-0.358	-0.270	-0.375
ΔG_{Vol-2} (eV)	-0.091 (-0.087 ^a)	-0.297	-0.114	-0.201
ΔG_{Hey} (eV)	-0.075 (-0.039 ^a)	0.197	0.270	0.375
ΔG_{Tafel} (eV)	0.297 (0.370 ^a)	0.578	1.006	1.128

The symbol of ^a indicates the reaction process of HER on another randomly active center of 5.9%-HEA (111) (see Supplementary Fig. 10).

Supplementary Fig. 10. Reaction process of HER on another random active center of 5.9%-HEA (111). (a) Volmer-Heyrovsky mechanism of HER on 5.9%-HEA (111). (b) Volmer-Tafel mechanism of HER on 5.9%-HEA (111). (c) The H^* spillover on DR1 (diffusion region for the first H^*) for 5.9%-HEA (111). (d) The H^* spillover on DR2 (diffusion region for the second H^*) for 5.9%-HEA (111).

Q2. In the revised paper, the sophisticated configuration of PtFeCoNiCu HEA system has been well investigated by randomly selecting more than 400 active sites. For clarity, the resulting ΔG_{H^*} values are recommended to be summarized and partially displayed in SI (for example, Fig. S55 and Table 9 in the article with DOI: 10.1088/2515-7655/abee33).

R2: We sincerely apologize for not finding Fig. S55 and Table 9 in the article with DOI: 10.1088/2515-7655/abee33. In response to the valuable suggestion provided by the reviewer, we have taken the necessary steps to address this issue. The ΔG_{H^*} values have been comprehensively summarized in the Supplementary Data section, accompanied by Supplementary Fig. 2. We greatly appreciate your feedback and thank you for bringing this to our attention.

“ ΔG_{H^*} was calculated on the designed HEA (111) with different strains (see Fig. 1d and Supplementary Fig. 2), including more than 400 datapoints (see Supplementary Data)” in lines 30-31 of page 3.

Supplementary Fig. 2. Adsorption free energy of H* (ΔG_{H^*}) on random active sites of PtFeCoNiCu HEA systems with different strains.

Report of Reviewer 4

Q. The authors answered all my questions. I recommend publishing the revised version of the manuscript.

R: Many thanks to the reviewer for the previous comments/suggestions, which significantly improved the quality of this manuscript.

REVIEWER COMMENTS

Reviewer #1 (Remarks to the Author):

I think the simulation of the compressed Pt(111) surface to fit the Pt(111) overlayers of the HEA surface, as well as the stretched HEA(111) surface to fit the lattice constant of unstrained Pt(111), is a valuable addition to the manuscript.

By inspection of Supplementary Figure 12 it appears clear that compression of a pure Pt(111) surface cannot solely be responsible for the increased activity at these low hydrogen coverages. The hydrogen adsorption thus appears to show further weakening on the HEA(111) surface beyond the compression of the Pt surface atomic layers. That alloyed elements a few layers under the compressed surface has such a substantial effect on the weakening of the adsorption energy is in itself a surprising result. In contrast to this, Supplementary Figure 13 shows what could be expected when the Pt overlayers have the same strain: the hydrogen adsorption energies of the strongest sites on the HEA and pure Pt are more or less identical. I find it peculiar that the ligand effect of alloyed elements displays such different effects in the two systems. In this regard, the authors would highly benefit from making their raw results (atomic geometries, energies, and simulation parameters; not merely a compilation of the energies) publicly available, so that there is full transparency regarding the presented conclusions. In Supplementary Figure 13 the transition state energy of the Tafel step on Pt is higher, consistent with Pt in Supplementary Figure 12. This decreased Tafel transition state energy of the HEA compared to Pt thus seems to me to be the most consistent difference between the HEA and Pt systems, and it is observed that the transition state energetics are favored in the HEA system. The reason for the favorable transition state energy in the HEA system could be many.

Now, regarding the interpretation of the results as an unusual Sabatier principle on which I remain unconvinced. I remain unconvinced of the truth of a phrasing by the authors like "It is noteworthy that for the 5.9%-HEA (111), all steps demonstrate an exothermic nature, indicating higher catalytic activity" in relation to Supplementary Figure 12a; but this holds true for all similar statements in the manuscript about the diffusion region being the reason for increased activity. I must acknowledge that the authors' and my premise for thinking about reaction rates differ too significantly to be reconciled. My counterargument to the above statement would be that such "exothermic" steps could always be found, also on pure Pt, but they are not determining for the reaction rate. An example is to consider that the bond of a hydrogen adsorbed to a surface can always be stretched (until the point of desorption as atomic hydrogen). The extent, however, of this vibration is, like the diffusion of the hydrogen on the surface, determined by the available thermal energy. The two situations (surface diffusion and bond stretching) would look identical in a free energy diagram, however.

I would acknowledge the reason for the increased activity of the HEA compared to Pt being the more classical reasons of weakening of the hydrogen bond (due to compressive strain and ligand effects), and decreased energies of the transition states. Asserting whether the spread in H adsorption energies would be a descriptor for these effects would require more compositions to be tested in experiments and simulations.

Detailed Responses to the Comments

Dear Reviewer 1,

We thank you for the favourable comments on our manuscript entitled “*Unusual Sabatier principle on high entropy alloy catalysts for hydrogen evolution reactions*” (manuscript No.: NCOMMS-23-13855B). We have considered these comments and addressed your concerns. Thus, we would like to submit the revised manuscript for your reconsideration.

Particularly, to address your comments, we have performed density functional theory calculation to reveal the role of H* spillover mechanism for the catalytic performance of HER, and additional experiments (XRD, electrochemical measurements, verification of H* spillover mechanism) to clarify the findings in this work, and have provided adequate experimental data, as well as all the raw results (atomic geometries, energies, and simulation parameters) to <https://github.com/chandrasinghuoft/PtFeCoNiCu.git>. In addition, 4 references have been added to detail and to strengthen the work in the revised manuscript.

Report of Reviewer 1

Q1 (question 1). *I think the simulation of the compressed Pt(111) surface to fit the Pt(111) overlayers of the HEA surface, as well as the stretched HEA(111) surface to fit the lattice constant of unstrained Pt(111), is a valuable addition to the manuscript.*

R1 (reply 1): The reviewer’s affirmation is greatly appreciated. We extend our sincere gratitude for previous comments from this reviewer, which have immensely contributed to enhancing the quality of this work.

Q2. *By inspection of Supplementary Figure 12 it appear clear that compression of a pure Pt(111) surface cannot solely be responsible for the increased activity at these low hydrogen coverages. The hydrogen adsorption thus appear to show further weakening on the HEA(111) surface beyond the compression of the Pt surface atomic layers. That alloyed elements a few layers under the compressed surface has such a substantial effect on the weakening of the adsorption energy is in itself a surprising result. In contrast to this, Supplementary Figure 13 shows what could be expected when the Pt overlayers have the same strain: the hydrogen adsorption energies of the strongest sites on the HEA and pure Pt are more or less identical. I find it peculiar that the ligand*

effect of alloyed elements displays such different effects in the two systems. In this regard, the authors would highly benefit from making their raw results (atomic geometries, energies, and simulation parameters; not merely a compilation of the energies) publicly available, so that there is full transparency regarding the presented conclusions. In Supplementary Figure 13 the transition state energy of the Tafel step on Pt is higher, consistent with Pt in Supplementary Figure 12. This decreased Tafel transition state energy of the HEA compared to Pt thus seem to me to be the most consistent difference between the HEA and Pt systems, and it is observed that the transition state energetics are favored in the HEA system. The reason for the favorable transition state energy in the HEA system could be many.

R2: We would like to express our thanks to the reviewer for pointing out this unusual phenomenon. We carefully examined the structure of H* adsorbed on Pt (111) and identified the strongest adsorption structure of HEA (111) for H*, as depicted in Fig. R1 (Please note that only the nearest and next-nearest neighbors are included). Among the 18 neighbors, only two exhibit dissimilarity. As a result, the ligand effect proves to be very weak in cases, where the adsorption site on HEA (111) is strong and the strain remains consistent. This suggests that the results presented in Supplementary Figure 13 make sense, aligning with the reviewer's perspective. Upon examining the side view of H* adsorbed on 5.9% Pt (111) (Fig. R2a), we observed some deformation in the Pt (111) surface. Subsequently, we removed the adsorbed H* and optimized the remaining Pt (111) structure. The final configuration of Pt (111) after the optimization, as illustrated in Fig. R2b, shows lower energy but significant deformation compared to the initial Pt (111). This deformation may be attributed to the large compression of 5.9%. Consequently, the corresponding DFT results for 5.9% Pt (111) appear implausible. In light of this, we have removed the relevant results (Supplementary Figure 12) and discussions from the revised manuscript. Fortunately, this adjustment does not impact our overall conclusion. The original supplementary Figure 13 still supports that the strain is only one of the reasons for the improved catalytic performance of HEAs.

To eliminate the effect of further weakening adsorption on 5.9%-HEA (111) beyond the compression of the Pt surface atomic layer, we constructed an idealized 5.9%-PtNi structure comprising one top layer of Pt and four bottom layers of Ni, resulting in a ΔG_{H^*} of 0.025 eV. This value is comparable to the ΔG_{H^*} (-0.099~0.075 eV) on 5.9%-HEA (111). The corresponding potential limiting step is the Volmer step with a ΔG_{Vol} of 0.025 eV and the energy barrier for H₂ formation is 0.612 eV, which is still higher than that (0.297 eV) on 5.9%-HEA (111), as illustrated

in Supplementary Fig. 13 (additional figure included in the supplementary materials). Such a DFT result suggests that, in addition to the weakened adsorption energy (due to the compressive strain and ligand effects), H* spillover plays a pivotal role in enhancing the catalytic activity. This has also been demonstrated in recent open literatures (Nat. Commun. 2022, 13, 5382; Nat. Commun. 2022, 13, 1189; Nat. Commun. 2021, 12, 3502; J. Am. Chem. Soc. 2022, 144, 6028-6039). Finally, following the reviewer's suggestion, we have uploaded all the raw results (atomic geometries, energies, and simulation parameters) to <https://github.com/chandrasinghuoft/PtFeCoNiCu.git> to enhance the clarity of our DFT results.

Fig. R1. (a) Adsorption structure of H* adsorbed on Pt (111). (b) The strongest adsorption site of HEA (111) for H*. Note that only the nearest and next-nearest neighbors are included.

Fig. R2. Geometrically optimized structures of (a) H* adsorbed on 5.9% Pt (111), (b) the remaining 5.9%-Pt (111) structure after removing H*, and (c) the initial 5.9%-Pt (111).

Supplementary Fig. 13. Reaction process of HER on 5.9%-HEA (111) and 5.9%-PtNi (111). (a) Volmer-Heyrovsky mechanism of HER on 5.9%-HEA (111) and 5.9%-PtNi (111). (b) Volmer-Tafel mechanism of HER on 5.9%-HEA (111) and 5.9%-PtNi (111).

The corresponding revisions in the main text:

“Furthermore, we constructed an idealized 5.9%-PtNi structure comprising one top layer of Pt and four bottom layers of Ni, resulting in a ΔG_{H^*} of 0.025 eV. This value is comparable to the ΔG_{H^*} (-0.099 ~ 0.075 eV) on 5.9%-HEA (111). The corresponding PLS is the Volmer step with a ΔG_{Vol} of 0.025 eV and the energy barrier for H_2 formation is 0.612 eV, which is still higher than that (0.297 eV) on 5.9%-HEA (111), as illustrated in Supplementary Fig. 13. Such a DFT result suggests that, in addition to the weakened adsorption energy (due to the compressive strain and ligand effects), H^* spillover plays a pivotal role in enhancing the catalytic activity. This has also been demonstrated in recent open literatures.^{20,22,23,34}” in lines 34-42 of page 5.

Q3. Now, regarding the interpretation of the results as an unusual Sabatier principle on which I remain unconvinced. I remain unconvinced of the truth of a phrasing by the authors like “It is noteworthy that for the 5.9%-HEA (111), all steps demonstrate an exothermic nature, indicating higher catalytic activity” in relation to Supplementary Figure 12a; but this holds true for all similar statements in the manuscript about the diffusion region being the reason for increased activity. I must acknowledge that the authors’ and my premise for thinking about reaction rates differ too significantly to be reconciled. My counterargument to the above statement would be that such “exothermic” steps could always be found, also on pure Pt, but they are not determining for the reaction rate. An example is to consider that the bond of a hydrogen adsorbed to a surface can always be stretched (until the point of desorption as atomic hydrogen). The extent, however, of this vibration is, like the diffusion of the hydrogen on the surface, determined by the available thermal energy. The two situations (surface diffusion and bond stretching) would look identical in a free

energy diagram, however.

I would acknowledge the reason for the increased activity of the HEA compared to Pt being the more classical reasons of weakening of the hydrogen bond (due to compressive strain and ligand effects), and decreased energies of the transition states. Asserting whether the spread in H adsorption energies would be a descriptor for these effects would require more compositions to be tested in experiments and simulations.

R3: Thanks to the reviewer for taking the time to provide valuable feedback on our manuscript. We appreciate the reviewer's thoughtful comments and the opportunity to address the reviewer's concerns.

Regarding the interpretation of our results, we understand the reviewer's reservations about the H* diffusion being the reason for increased activity. First, in our work, we harnessed the high entropy properties of HEA to create a diverse array of adsorption sites, enabling H* spillover, a phenomenon substantiated by a comprehensive series of experimental results. This is also consistent with the reviewer's viewpoint in the second-round review. Moreover, we have delved into the existing literatures and found that the role of H* spillover in enhancing catalytic performance of HER has been reported in previous studies (Nat. Commun. 2022, 13, 5382; Nat. Commun. 2022, 13, 1189; Nat. Commun. 2021, 12, 3502; J. Am. Chem. Soc. 2022, 144, 6028-6039). Notably, Shao et al. reported a single-phase complex oxide $\text{La}_2\text{Sr}_2\text{PtO}_{7+\delta}$ as a high-performance hydrogen evolution electrocatalyst in acidic media utilizing an atomic-scale H* spillover effect between multifunctional catalytic sites (Nat. Commun. 2022, 13, 1189). The schematic illustration of the H* spillover on two-type catalyst systems for HER in acidic media is presented in Fig. R3a-b. Furthermore, Ma et al. identified that a small work function difference induces interfacial charge dilution and relocation, thereby weakening interfacial proton adsorption and enabling efficient H* spillover for HER (Nat. Commun. 2021, 12, 3502, as shown in Fig. R3c). These studies collectively emphasize that H* spillover can significantly promote HER. In their proposed H* spillover mechanism, the adsorption of H* on different sites (strong, moderate, and weak sites) should exhibit stable or metastable morphology, which is a little different from the vibrational behavior of the bond.

To further substantiate the role of H* spillover in promoting HER within our system, we have conducted additional experiments (additional Supplementary Fig. 50 included in the supplementary materials) and DFT calculations. First, a PtNi₃ alloy was synthesized with the

corresponding XRD pattern depicted in Supplementary Fig. 50a. To detect the H* spillover mechanism on PtNi₃, a color change experiment was conducted. WO₃ was mixed with PtNi₃ and the mixture of PtNi₃@WO₃ went through a HER process for 30 min (see Supplementary Fig. 50b). After the HER process, the color of PtNi₃@WO₃ remained unchanged (see Supplementary Fig. 50c), indicating that no H* spillover mechanism occurs in the PtNi₃ system (Nat. Commun. 2022, 13, 1189). As a result, the PtNi₃ catalyst (31.2 mV @ -10 mA cm⁻²) shows inferior HER performance to that of the designed HEA-400-5000 (10.8 mV @ -10 mA cm⁻², see Supplementary Fig. 50d), even though the ΔG_{H^*} value of PtNi₃ may be in close proximity to that of HEA-400-5000. Furthermore, by DFT calculation, we have determined the ΔG_{H^*} value on 5.9%-PtNi (111) to be 0.025 eV, falling within the range of ΔG_{H^*} (-0.099 ~ 0.075 eV) on 5.9%-HEA (111). As discussed in response to the second question, our DFT result suggests that, alongside the weakened adsorption energy resulting from compressive strain and ligand effects, the spillover of H* plays a pivotal role in enhancing the catalytic activity. This conclusion is consistent with other reports in open literatures (Nat. Commun. 2022, 13, 5382; Nat. Commun. 2022, 13, 1189; Nat. Commun. 2021, 12, 3502; J. Am. Chem. Soc. 2022, 144, 6028-6039). From the above discussions, both the experimental results and DFT calculations affirm the contribution of the H* spillover mechanism to the improved catalytic performance of HER. We have revised the statements concerning the diffusion region and its impact on catalytic activity, with the aim of providing a clearer and more accurate representation.

Fig. R3. (a) The conventional hydrogen spillover-based binary-component catalyst system by coupling hydrogen-enriched Pt-based nanocrystals with hydrogen-deficient component. Red balls represent Pt atoms and blue and gray balls represent compounds. (b) Hydrogen spillover-based single-component catalyst system with atomic level multiple catalytic sites. Red, blue and gray balls represent strong H adsorption, thermoneutral H adsorption and facile H₂ desorption sites, respectively. (c) Proposed nature of the work functional difference ($\Delta\Phi$) on the H* spillover phenomenon in hydrogen spillover-based binary catalysts. (a-b) Reprinted with permission from Nat. Commun. 2022, 13, 1189; (c) Reprinted with permission from Nat. Commun. 2021, 12, 3502.

Supplementary Fig. 50. (a) XRD patterns of PtNi₃. (b) HER galvanostatic plots for mixture of PtNi₃@WO₃ to identify the spillover mechanism. (c) Color change photographs of PtNi₃@WO₃. The photographs are taken before and after the HER process as shown in Supplementary Fig. 50b. (d) Polarization curves of HEA-400-5000 and PtNi₃.

The corresponding revisions in the main text:

“Another potential phenomenon that could circumvent the volcano relationship is H* spillover, in which H* exhibits the ability to diffuse between different active sites.²⁰⁻²² Dai *et al.* reported a single-phase complex oxide La₂Sr₂PtO_{7+δ} as a high performance HER electrocatalyst utilizing an atomic-scale H* spillover effect between multifunctional catalytic sites.²³” in lines 22-25 of page 2.

“With the consideration of H* spillover, all the electrochemical steps become spontaneous reactions, and the RDS manifests as the H* spillover on the surface of 5.9%-HEA (111) with the largest energy barrier of 0.124 eV.” in lines 12-14 of page 5.

“Additionally, the HER performance of HEA-400-5000 was also compared with a PtNi₃ system without H* spillover effect, as shown in Supplementary Fig. 50. HEA-400-5000 shows better HER performance than PtNi₃ (31.2 mV at -10 mA cm⁻²), even though the ΔG_{H*} value of PtNi₃ may be in close proximity to that of HEA-400-5000 (ΔG_{H*} in range around -0.099 ~ 0.075 eV), indicating the spillover of H* plays a pivotal role in enhancing the catalytic activity.” in lines 14-18 of page

9.

“The synthesis of PtNi₃ was using the method according to the report of Wang *et al.*⁴⁴ Typically, 0.4 mmol of H₂PtCl₆·6H₂O, 1.2 mmol Ni(NO₃)₂·6H₂O and 287.2 mg polyvinyl pyrrolidone (PVP) were dissolved into 50 mL ultrapure water and sonicated for 1 h. Then, the solution was sprayed onto a glass plate maintained at 400 °C for rapid evaporation. The collected powder was then cleaned through centrifugation for 3 times with ultrapure water. The PtNi₃ were obtained through annealing the powder under 5% H₂/Ar atmosphere for 2 h at 500 °C.” in lines 1-6 of page 11.

REVIEWERS' COMMENTS

Reviewer #1 (Remarks to the Author):

I will acknowledge the authors that they, intentionally or unintentionally, have shifted my stand on the relation of the hydrogen spillover to the notion of conservative forces by a nudge: The situation of “metastable morphology” of “the adsorption of H^* on different sites” that the authors refer to in their rebuttal letter, surely is distinct from the H^* vibration I was referring to. They are still comparable in the sense that the “diffusion region” would look identical to the stretching of the H^* surface bond in a free energy diagram, because they both can be described as particles moving under the influence of conservative forces. However, it seems clearer to me now that the metastabilities involved in the diffusion region indeed become relevant when the forces are not conservative, as is the case when thermal energy is involved. The H^* might indeed rest kinetically trapped in a metastable state until the surface has had time to “heat up” (by adsorbing heat from the surroundings) providing the H^* more thermal energy to continue its hopping on the surface. Surely the reference to conservative forces breaks down when this is the assumed physical description, whereby also the assumptions in transition state theory are relaxed.

As such, the hydrogen evolution activity could duly be governed by an unusual Sabatier principle.

The structure geometries and energies shared by the authors is an excellent first step in increasing the transparency of the stated conclusions. I was intrigued by the opportunity to have a look at the structures and energies related to the Tafel transition states of Pt(111) and the 5.9%-HEA in Figure 2c. However, I was not successful in figuring out which of the many configurations of the structures scattered over multiple files have made the basis for the results presented in Figure 2 in the hour or so that I allocated to the attempt. I will encourage the authors to provide a script in their repository that would show how their presented results are obtained, to improve ease of transparency.

The reason I am particularly interested in the ease of transparency in relation to comparing the transition states is that this will provide the readers the opportunity to test, perhaps falsify, the hypothesis that an unusual Sabatier principle is the reason for the decreased transition state energy (the location of the transition state energy level, not the barrier height) of the Tafel step on the HEA, which to me still is impossible to explain with an unusual Sabatier principle, since this principle is only related to the surface diffusion. An example of a manifestation of this can be found by inspection of Figure 6d in DOI: 10.1038/s41467-022-28843-2 by Dai et al., a paper that the authors also refer to. There, the transition state energies, albeit for the Heyrovsky step and not the Tafel step as in the current work, are approximately identical, independent of the H^* migration. Nonetheless the HEA system of the current work appears to show a decreased transition state energy, that likely has an important contribution to the catalytic activity, but the reason for this is to me still unaccounted for. I acknowledge that this, however, may be out of the scope of the current work.

Detailed Responses to the Comments

Reviewer 1: *I will acknowledge the authors that they, intentionally or unintentionally, have shifted my stand on the relation of the hydrogen spillover to the notion of conservative forces by a nudge: The situation of “metastable morphology” of “the adsorption of H* on different sites” that the authors refer to in their rebuttal letter, surely is distinct from the H* vibration I was referring to. They are still comparable in the sense that the “diffusion region” would look identical to the stretching of the H* surface bond in a free energy diagram, because they both can be described as particles moving under the influence of conservative forces. However, it seems clearer to me now that the metastabilities involved in the diffusion region indeed become relevant when the forces are not conservative, as is the case when thermal energy is involved. The H* might indeed rest kinetically trapped in a metastable state until the surface has had time to “heat up” (by adsorbing heat from the surroundings) providing the H* more thermal energy to continue its hopping on the surface. Surely the reference to conservative forces breaks down when this is the assumed physical description, whereby also the assumptions in transition state theory are relaxed. As such, the hydrogen evolution activity could duly be governed by an unusual Sabatier principle.*

The structure geometries and energies shared by the authors is an excellent first step in increasing the transparency of the stated conclusions. I was intrigued by the opportunity to have a look at the structures and energies related to the Tafel transition states of Pt(111) and the 5.9%-HEA in Figure 2c. However, I was not successful in figuring out which of the many configurations of the structures scattered over multiple files have made the basis for the results presented in Figure 2 in the hour or so that I allocated to the attempt. I will encourage the authors to provide a script in their repository that would show how their presented results are obtained, to improve ease of transparency.

The reason I am particularly interested in the ease of transparency in relation to comparing the transition states is that this will provide the readers the opportunity to test, perhaps falsify, the hypothesis that an unusual Sabatier principle is the reason for the decreased transition state energy (the location of the transition state energy level, not the barrier height) of the Tafel step on the HEA, which to me still is impossible to explain with an unusual Sabatier principle, since this principle is only related to the surface diffusion. An example of a manifestation of this can be found by inspection of Figure 6d in DOI: 10.1038/s41467-022-28843-2 by Dai et al., a paper that the authors also refer to. There, the transition state energies, albeit for the Heyrovsky step and not the Tafel step as in the current work, are approximately identical, independent of the H migration. Nonetheless the HEA system of the current work appears to show a decreased transition state energy, that likely has an important contribution to the catalytic activity, but the reason for this is to me still unaccounted for. I acknowledge that this, however, may be out of the scope of the current work.*

Reply: Thank you sincerely for the thorough and insightful feedback provided. We are genuinely grateful for the reviewer's clarification concerning the 'metastable morphology' of H* adsorption and the unusual Sabatier principle governing hydrogen evolution activity. We acknowledge your concern about the decreased transition state energy of the Tafel step on the HEA. In reference to the example you provided from Dai et al.'s work (Ref. 23 in the manuscript), we draw attention to our own results (Fig. 2c and Supplementary Fig. 11b) that manifest a comparable phenomenon. While the position of the transition state energy level aligns closely, the distinction lies in the energy of 2H* (reactants), leading to disparate energy barriers. We appreciate your recognition

that the reason for the decreased transition state energy might extend beyond the current scope of our work. We will devote more effort on this issue in future research.

We apologize for any confusion caused by the multiple and unclear files. In response to this concern and with a shared commitment to improving transparency, we wholeheartedly embrace your suggestion to furnish a more detailed description within our repository (<https://github.com/chandrasinghuoft/PtFeCoNiCu.git>). This enhancement encompasses not only the theoretical calculation data but also the experimental data, meticulously organized to eradicate ambiguity. We are committed to enhancing the accessibility of our data. Thank you once again for your constructive input, and we are committed to making the necessary revisions to meet the highest standards of clarity and reproducibility.